# Modular Continual Learning in a Unified Visual Environment

**Kevin T. Feigelis**
Department of Physics
Stanford Neurosciences Institute
Stanford University
Stanford, CA 94305
feigelis@stanford.edu

**Blue Sheffer**
Department of Psychology
Stanford University
Stanford, CA 94305
bsheffer@stanford.edu

**Daniel L. K. Yamins**
Departments of Psychology and Computer Science
Stanford Neurosciences Institute
Stanford University
Stanford, CA 94305
yamins@stanford.edu

## Abstract

A core aspect of human intelligence is the ability to learn new tasks quickly and switch between them flexibly. Here, we describe a modular continual reinforcement learning paradigm inspired by these abilities. We first introduce a visual interaction environment that allows many types of tasks to be unified in a single framework. We then describe a reward map prediction scheme that learns new tasks robustly in the very large state and action spaces required by such an environment. We investigate how properties of module architecture influence efficiency of task learning, showing that a module motif incorporating specific design principles (e.g. early bottlenecks, low-order polynomial nonlinearities, and symmetry) significantly outperforms more standard neural network motifs, needing fewer training examples and fewer neurons to achieve high levels of performance. Finally, we present a meta-controller architecture for task switching based on a dynamic neural voting scheme, which allows new modules to use information learned from previously-seen tasks to substantially improve their own learning efficiency.

## Introduction

In the course of everyday functioning, people are constantly faced with real-world environments in which they are required to shift unpredictably between multiple, sometimes unfamiliar, tasks (Botvinick & Cohen, 2014). They are nonetheless able to flexibly adapt existing decision schemas or build new ones in response to these challenges (Arbib, 1992). How humans support such flexible learning and task switching is largely unknown, both neuroscientifically and algorithmically (Wagner et al., 1998; Cole et al., 2013).

We investigate solving this problem with a *neural module* approach in which simple, task-specialized decision modules are dynamically allocated on top of a largely-fixed underlying sensory system (Andreas et al., 2015; Hu et al., 2017). The sensory system computes a general-purpose visual representation from which the decision modules read. While this sensory backbone can be large, complex, and learned comparatively slowly with significant amounts of training data, the task modules that deploy information from the base representation must, in contrast, be lightweight, quick to be learned, and easy to switch between. In the case of visually-driven tasks, results from neuroscience and computer vision suggest the role of the fixed general purpose visual representation may be played by the ventral visual stream, modeled as a deep convolutional neural network (Yamins & DiCarlo, 2016; Razavian et al., 2014). However, the algorithmic basis for how to efficiently learn and dynamically deploy visual decision modules remains far from obvious.

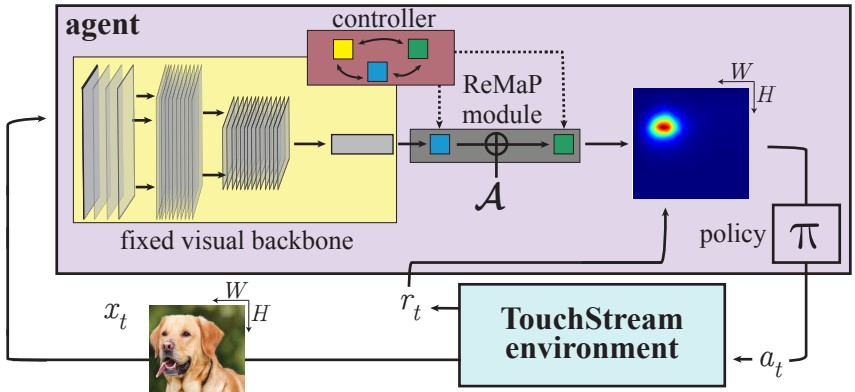

Figure 1: **Modular continual learning in the TouchStream environment** The TouchStream environment is a touchscreen-like GUI for continual learning agents, in which a spectrum of visual reasoning tasks can be posed in a large but unified action space. On each timestep, the environment (cyan box) emits a visual image ($x_t$) and a reward ($r_t$). The agent recieves $x_t$ and $r_t$ as input and emits an action $a_t$. The action represents a "touch" at some location on a two-dimensional screen e.g. $a_t \in \{0, \ldots, H-1\} \times \{0, \ldots, W-1\}$, where $H$ and $W$ are the screen height and width. The environment's policy is a program computing $x_t$ and $r_t$ as a function of the agent's action history. The agent's goal is to learn how to choose optimal actions to maximize the amount of reward it recieves over time. The agent consists of several component neural networks including a fixed visual backbone (yellow inset), a set of learned neural modules (grey inset), and a meta-controller (red inset) which mediates the deployment of these learned modules for task solving. The modules use the ReMaP algorithm § 2 to learn how to estimate reward as a function of action (heatmap), conditional on the agent's recent history. Using a sampling policy on this reward map, the agent chooses an optimal action to maximize its aggregate reward.

In standard supervised learning, it is often assumed that the output space of a problem is prespecified in a manner that just happens to fit the task at hand – e.g. for a classification task, a discrete output with a fixed number of classes might be determined ahead of time, while for a continuous estimation problem, a one-dimensional real-valued target might be chosen instead. This is a very convenient simplification in supervised learning or single-task reinforcement learning contexts, but if one is interested in the learning and deployment of decision structures in a rich environment defining tasks with many different natural output types, this simplification becomes cumbersome.

To go beyond this limitation, we build a *unified environment* in which many different tasks are naturally embodied. Specifically, we model an agent interacting with a two-dimensional touchscreen-like GUI that we call the *TouchStream*, in which all tasks (discrete categorization tasks, continuous estimation problems, and many other combinations and variants thereof) can be encoded using a single common and intuitive – albeit large – output space. This choice frees us from having to hand-design or programmatically choose between different output domain spaces, but forces us to confront the core challenge of how a naive agent can quickly and emergently learn the implicit "interfaces" required to solve different tasks.

We then introduce Reward Map Prediction (ReMaP) networks, an algorithm for continual reinforcement learning that is able to discover implicit task-specific interfaces in large action spaces like those of the TouchStream environment. We address two major algorithmic challenges associated with learning ReMaP modules. First, what module architectural motifs allow for *efficient* task interface learning? We compare several candidate architectures and show that those incorporating certain intuitive design principles (e.g. early visual bottlenecks, low-order polynomial nonlinearities and symmetry-inducing concatenations) significantly outperform more standard neural network motifs, needing fewer training examples and fewer neurons to achieve high levels of performance. Second, what system architectures are effective for *switching* between tasks? We present a meta-controller architecture based on a dynamic neural voting scheme, allowing new modules to use information learned from previously-seen tasks to substantially improve their own learning efficiency.

In § 1 we formalize the TouchStream environment. In § 2, we introduce the ReMaP algorithm. In § 3, we describe and evaluate comparative performance of multiple ReMaP module architectures on a variety of TouchStream tasks. In § 4, we describe the Dynamic Neural Voting meta-controller, and evaluate its ability to efficiently transfer knowledge between ReMaP modules on task switches.

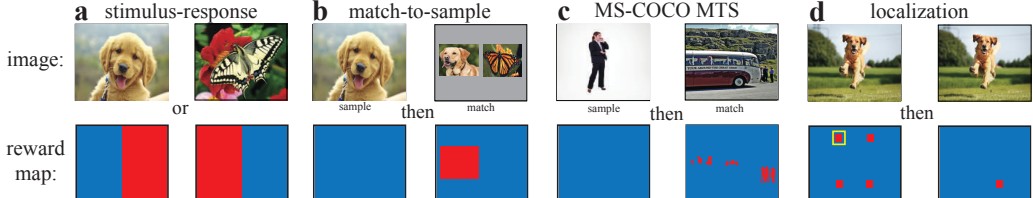

Figure 2: **Exemplar TouchStream tasks.** Illustration of several task paradigms explored in this work using the TouchStream Environment. The top row depicts observation $x_t$ and the bottom shows the ground truth reward maps (with red indicating high reward and blue indicating low reward). **a.** Binary Stimulus-Response task. **b.** stereotyped Match-To-Sample task. **c.** The Match-To-Sample task using the MS-COCO dataset. **d.** Object localization.

## RELATED WORK

Modern deep convolutional neural networks have had significant impact on computer vision and artificial intelligence (Krizhevsky et al., 2012), as well as in the computational neuroscience of vision (Yamins & DiCarlo (2016)). There is a recent but growing literature on convnet-based *neural modules*, where they have been used for solving compositional visual reasoning tasks (Andreas et al., 2015; Hu et al., 2017). In this work we apply the idea of modules to solving visual learning challenges in a continual learning context. Existing works rely on choosing between a menu of pre-specified module primitives, using different module types to solve subproblems involving specific input-output datatypes, without addressing *how* these modules' forms are to be discovered in the first place. In this paper, we show a single generic module architecture is capable of automatically learning to solve a wide variety of different tasks in a unified action/state space, and a simple controller scheme is able to switch between such modules.

Our results are also closely connected with the literature on lifelong (or continual) learning (Kirkpatrick et al., 2016; Rusu et al., 2016). A part of this literature is concerned with learning to solve new tasks without catastrophically forgetting how to solve old ones (Zenke et al., 2017; Kirkpatrick et al., 2016). The use of modules obviates this problem, but instead shifts the hard question to one of how newly-allocated modules can be learned effectively. The continual learning literature also directly addresses knowlege transfer to newly allocated structures (Chen et al., 2015; Rusu et al., 2016; Fernando et al., 2017), but largely addresses how transfer learning can lead to higher performance, rather than addressing how it can improve learning speed. Aside from reward performance, we focus on issues of speed in learning and task switching, motivated by the remarkably efficient adaptability of humans in new task contexts. Existing work in continual learning also largely does not address which specific architecture types learn tasks efficiently, independent of transfer. By focusing first on identifying architectures that achieve high performance quickly on individual tasks (§ 3), our transfer-learning investigation then naturally focuses more on how to efficiently identify when and how to re-use components of these architectures (§ 4). Most of these works also make explicit *a priori* assumptions about the structure of the tasks to be encoded into the models (e.g. output type, number of classes), rather than address the more general question of emergence of solutions in an embodied case, as we do.

Meta-reinforcement learning approaches such as Wang et al. (2016); Duan et al. (2016), as well as the schema learning ideas of e.g. Arbib (1992); McClelland (2013) typically seek to address the issue of continual learning by having a complex meta-learner extract correlations between tasks over a long timescale. In our context most of the burden of environment learning is placed on the individual modules, so our meta-controller can thus be comparatively light-weight compared to typical meta-reinforcement approaches. Unlike our case, meta-learning has mostly been limited to small state or action spaces. Some recent work in general reinforcement learning (e.g. Ostrovski et al. (2017); Dulac-Arnold et al. (2015)) has addressed the issue of large action spaces, but has not sought to address multitask transfer learning in these large action spaces.

## 1 THE TOUCHSTREAM ENVIRONMENT

Agents in a real-world environment are exposed to many different implicit tasks, arising without predefined decision structures, and must learn *on the fly* what the appropriate decision interfaces are

for each situation. Because we are interested in modeling how agents can do this on-the-fly learning, our task environment should mimic the unconstrained nature of the real world. Here, we describe the TouchStream environment, which attempts to do this in a simplified two-dimensional domain.

Our problem setup consists of two components, an "environment" and an "agent," interacting over an extended temporal sequence (Fig. 1). At each timestep $t$, the environment emits an RGB image $x_t$ of height $H$ and width $W$, and a scalar reward $r_t$. Conversely, the agent accepts images and rewards as input and chooses an action $a_t$ in response. The action space $\mathcal{A}$ available to the agent consists of a two-dimensional pixel grid $\{0, \ldots, H-1\} \times \{0, \ldots, W-1\} \subset \mathbb{Z}^2$, of the same height and width as its input image. The environment is equipped with a policy (unknown to the agent) that on each time step computes image $x_t$ and reward $r_t$ as a function of the history of agent actions $\{a_0, \ldots, a_{t-1}\}$, images $\{x_0, \ldots, x_{t-1}\}$ and rewards $\{r_0, \ldots, r_{t-1}\}$.

In this work, the agent is a neural network, composed of a visual backbone with fixed weights, together with a meta-controller module whose parameters are learned by interaction with the environment. The agent's goal is to learn to enact a policy that maximizes its reward obtained over time. Unlike an episodic reinforcement learning context, the TouchStream environment is continuous: throughout the course of learning the agent is never signaled when it should reset to some "initial" internal state. However, unlike the traditional continuous learning context of e.g. Sutton & Barto (1998), a TouchStream may implicitly define many different tasks, each of which is associated with its own characteristic reward schedule. The agent experiences a continual stream of tasks, and any implicit association between reward schedule and state reset must be discovered by the agent.

By framing the action space $\mathcal{A}$ of the agent as all possible pixel locations and the state space as any arbitrary image, a very wide range of possible tasks are unified in this single framework, at the cost of requiring the agents' action space to be congruent to its input state space, and thus be quite large. This presents two core efficiency challenges for the agent: on any given task, it must be able to both quickly recognize what the "interface" for the task is, and *transfer* such knowledge across tasks in a smart way. Both of these goals are complicated by the fact that both the large size of agent's state and action spaces.

Although we work with modern large-scale computer vision-style datasets and tasks in this work, e.g. ImageNet (Deng et al. (2009)) and MS-COCO (Lin et al. (2014)), we are also inspired by visual psychology and neuroscience, which have pioneered techniques for how controlled visual tasks can be embodied in real reinforcement learning paradigms (Horner et al., 2013; Rajalingham et al., 2015). Especially useful are three classes of task paradigms that span a range of the ways discrete and continuous estimation tasks can be formulated – including Stimulus-Response, Match-To-Sample, and Localization tasks (Fig. 2).

**Stimulus-Response Tasks:** The Stimulus-Response (SR) paradigm is a common approach to physically embodying discrete categorization tasks (Gaffan & Harrison, 1988). For example, in the simple two-way SR discrimination task shown in Fig. 2a, the agent is rewarded if it touches the left half of the screen after being shown an image of a dog, and the right half after being shown a butterfly. SR tasks can be made more difficult by increasing the number of image classes or the complexity of the reward boundary regions. In our SR experiments, we use images and classes from the ImageNet dataset (Deng et al., 2009).

**Match-To-Sample Tasks:** The Match-to-Sample (MTS) paradigm is another common approach to assessing visual categorization abilities (Murray & Mishkin, 1998). In the MTS task shown in Fig. 2b, trials consist of a sequence of two image frames – the "sample" screen followed by the "match" screen – in which the agent is expected to remember the object category seen on the sample frame, and then select an onscreen "button" (really, a patch of pixels) on the match screen corresponding to the sample screen category. Unlike SR tasks, MTS tasks require some working memory and more localized spatial control. More complex MTS tasks involve more sophisticated relationships between the sample and match screen. In Fig. 2c, using the MS-COCO object detection challenge dataset (Lin et al., 2014), the sample screen shows an isolated template image indicating one of the 80 MS-COCO classes, while the match screen shows a randomly-drawn scene from the dataset containing at least one instance of the sample-image class. The agent is rewarded if its chosen action is located inside the boundary of an instance (e.g. the agent "pokes inside") of the correct class. This MS-COCO MTS task is a "hybrid" of categorical and continuous elements, meaning that if phrased as a standard

supervised learning problem, both categorical readout (i.e. class identity) and a continous readout (i.e. object location) would be required.

**Localization:** Fig. 2d shows a two-step continuous localization task in which the agent is supposed to mark out the bounding box of an object by touching opposite corners on two successive timesteps, with reward proportionate to the Intersection over Union (IoU) value of the predicted bounding box relative to the ground truth bounding box $IoU = \frac{Area(B_{GT} \cap \hat{B})}{Area(B_{GT} \cup \hat{B})}$. In localization, unlike the SR and MTS paradigms, the choice made at one timestep constrains the agent's optimal choice on a future timestep (e.g. picking the upper left corner of the bounding box on the first step contrains the lower right opposite corner to be chosen on the second).

Although these tasks can become arbitrarily complex along certain axes, the tasks presented here require only fixed-length memory and future prediction. That is, each task requires only knowledge of the past $k_b$ timesteps, and a perfect solution always exists within $k_f$ timesteps from any point. The minimal required values of $k_b$ and $k_f$ are different across the various tasks in this work. However, in the investigations below, we set these to the maximum required values across tasks, i.e. kb = 1 and kf = 2. Thus, the agent is required to learn for itself when it is safe to ignore information from the past and when it is irrelevant to predict past a certain point in the future.

We will begin by considering a restricted case where the environment runs one semantic task indefinitely, showing how different architectures learn to solve such individual tasks with dramatically different levels of efficiency (§ 2-3). We will then expand to considering the case where the environment's policy consists of a sequence of tasks with unpredictable transitions between tasks, and exhibit a meta-controller that can cope effectively with this expanded domain (§ 4).

## 2 REWARD MAP PREDICTION

The TouchStream environment necessarily involves working with large action and state spaces. Methods for handling this situation often focus on reducing the effective size of action/state spaces, either via estimating pseudo-counts of state-action pairs, or by clustering actions (Ostrovski et al., 2017; Dulac-Arnold et al., 2015). Here we take another approach, using a neural network to directly approximate the (image-state modulated) mapping between the action space and reward space, allowing learnable regularities in the state-action interaction to implicitly reduce the large spaces into something manageable by simple choice policies. We introduce an off-policy algorithm for efficient multitask reinforcement learning in large action and state spaces: Reward Map Prediction, or ReMaP.

### 2.1 ReMaP NETWORK ALGORITHM

As with any standard reinforcement learning situation, the agent seeks to learn an optimal policy $\pi = p(a_t \mid x_t)$ defining the probability density $p$ over actions given image state $x_t$. The ReMaP algorithm is off-policy, in that $\pi$ is calculated as a simple fixed function of the estimated reward.

A *ReMaP network* $M_\Theta$ is a neural network with parameters $\Theta$, whose inputs are a history over previous timesteps of (i) the agent's own actions, and (ii) an activation encoding of the agent's state space; and which explicitly approximates the expected reward map across its action space for some number of future timesteps. Mathematically:

$$M_\Theta : \left[ \boldsymbol{\Psi}_{t-k_b:t}, \mathbf{h}_{t-k_b:t-1} \right] \longmapsto \left[ m_t^1, m_t^2, \ldots, m_t^{k_f} \right]$$

where $k_b$ is the number of previous timesteps considered; $k_f$ is the length of future horizon to be considered; $\boldsymbol{\Psi}_{t-k_b:t}$ is the history $[\psi(x_{t-k_b}), \ldots, \psi(x_t)]$ of state space encodings produced by fixed backbone network $\psi(\cdot)$, $\mathbf{h}_{t-k_b:t-1}$ is the history $[a_{t-k_b} \ldots, a_{t-1}]$ of previously chosen actions, and each $m_i \in \mathbf{map}(\mathcal{A}, \mathcal{R})$ – that is, a map from action space to reward space. The predicted reward maps are constructed by computing the expected reward obtained for a subsample of actions drawn randomly from $\mathcal{A}$:

$$m_t^j : a_t \mapsto E\left[ r_{t+j} \mid a_t, \mathbf{h}_{t-k_b:t-1}, \boldsymbol{\Psi}_{t-k_b:t} \right] = \int_{\mathcal{R}} r_{t+j} p(r_{t+j} \mid a_t, \mathbf{h}_{t-k_b:t-1}, \boldsymbol{\Psi}_{t-k_b:t}). \quad (1)$$

where $r_{t+j}$ is the predicted reward $j$ steps into the future horizon. Having produced $k_f$ reward prediction maps, one for each timestep of its future horizon, the agent needs to determine what

it believes will be the single best action over all the expected reward maps $\left[m_t^1, m_t^2, \ldots, m_t^{k_f}\right]$. The ReMaP algorithm formulates doing so by normalizing the predictions across each of these $k_f$ maps into separate probability distributions, and sampling an action from the distribution which has maximum variance. That is, the agent computes its policy $\pi$ as follows:

$$\pi = \mathbf{VarArgmax}_{j=1}^{k_f}\{\mathbf{Dist}[\mathbf{Norm}[m_t^j]]\}, \tag{2}$$

where

$$Norm[m] = m - \min_{x \in \mathcal{A}} m(x) \tag{3}$$

is a normalization that removes the minimum of the map,

$$Dist[m] = \frac{f(m)}{\int_{\mathcal{A}} f(m(x))} \tag{4}$$

ensures it is a probability distribution parameterized by functional family $f(\cdot)$, and $\mathbf{VarArgmax}$ is an operator which chooses the input with largest variance.

The sampling procedure described in equation (2) uses two complementary ideas to exploit spatial and temporal structure to efficiently explore a large action space. Since rewards in real physical tasks are spatially correlated, the distribution-based sampler in Equation (4) allows for more effective exploration of potentially informative actions than would the single-point estimate of an apparent optimum (e.g. an $\epsilon$-greedy policy). Further, in order to reduce uncertainty, the ReMaP algorithm explores timesteps with greatest reward map variance. The VarArgmax function nonlinearly upweights the timeframe with highest variance to exploit the fact that some points in time carry disproportionate relevance for reward outcome, somewhat analagously to how max-pooling operates in convolutional networks. Although any standard action selection strategy can be used in place of the one in (2) (e.g. pseudo $\epsilon$-greedy over all $k_f$ maps), we have empirically found that this policy is effective at efficiently exploring our large action space.

The parameters $\Theta$ of a ReMaP network are learned by gradient descent on the loss of the reward prediction error $\Theta^* = \mathrm{argmin}_\Theta L\left[m_t(a_t), r_t, ; \Theta\right]$ with map $m_t^j$ compared to the true reward $r_{t+j}$. Only the reward prediction in $m_t$ corresponding to the action chosen at timestep $t$ participates in loss calculation and backpropagation of error signals. A minibatch of maps, rewards, and actions is collected over several consecutive inference passes before performing a parameter update.

The ReMaP algorithm is summarized in 1.

---

**Algorithm 1: ReMaP – Reward Map Prediction**

---

Initialize ReMaP network $M$
Initialize state and action memory buffers $\mathbf{\Psi}_{t-k_b:t}$ and $\mathbf{h}_{t-k_b:t-1}$
**for** *timestep t = 1,T* **do**
    Observe $x_t$, encode with state space network $\psi(\cdot)$, and append to state buffer
    Subsample set of potential action choices $a_t$ uniformly from $\mathcal{A}$
    Produce $k_f$ expected reward maps of $a_t$ from eq. (1)
    Select action according to policy $\pi$ as in (2)
    Execute action $a_t$ in environment, store in action buffer, and receive reward $r_t$
    Calculate loss for this and previous $k_f - 1$ timesteps
    **if** $t \equiv 0 \mod$ *batch size* **then**
        Perform parameter update

---

Throughout this work, we take our fixed backbone state space encoder to be the VGG-16 convnet, pretrained on ImageNet (Simonyan & Zisserman, 2014). Because the resolution of the input to this network is 224x224 pixels, our action space $\mathcal{A} = \{0, \ldots, 223\} \times \{0, \ldots, 223\}$. By default, the functional family $f$ used in the action selection scheme in Eq. (4) is the identity, although on tasks benefiting from high action precision (e.g. Localization or MS-COCO MTS), it is often optimal to sample a low-temperature Boltzmann distribution with $f(x) = e^{-x/T}$. Reward prediction errors are calculated using the cross-entropy loss (where logits are smooth approximations to the Heaviside function in analogy to eq. (5)).

## 3 EFFICIENT NEURAL MODULES FOR TASK LEARNING

The main question we seek to address in this section is: what specific neural network structure(s) should be used in ReMaP modules? The key considerations are that such modules (i) should be easy to learn, requiring comparatively few training examples to discover optimal parameters $\Theta^*$, and (ii) easy to learn from, meaning that an agent can quickly build a new module by reusing components of old ones.

**Intuitive Example:** As an intuition-building example, consider the case of a simple binary Stimulus-Response task, as in Fig. 2a ("if you see a dog touch on the right, if a butterfly touch on the left"). One decision module that is a "perfect" reward predictor on this task is expressed analytically as:

$$M[\mathbf{\Psi}_t](a_x, a_y) = H(\mathbf{ReLU}(W\mathbf{\Psi}_t) \cdot \mathbf{ReLU}(a_x) + \mathbf{ReLU}(-W\mathbf{\Psi}_t) \cdot \mathbf{ReLU}(-a_x)) \quad (5)$$

where $H$ is the Heaviside function, $a_x$ and $a_y$ are the $x$ and $y$ components of the action $a \in \mathcal{A}$ relative to the center of the screen, and $W$ is a $1 \times |\mathbf{\Psi}_t|$ matrix expressing the class boundary (bias term omitted for clarity). If $W\mathbf{\Psi}_t$ is positive (i.e. the image is of a dog) then $a_x$ must also be positive (i.e. touch is on the right) to predict positive reward; conversly, if $W\mathbf{\Psi}_t$ is negative (i.e. butterfly), $a_x$ must be negative (i.e. left touch) to predict reward. If neither of these conditions hold, both terms are equal to zero, so the formula predicts no reward. Since vertical location of the action does not affect reward, $a_y$ is not involved in reward calculation on this task.

Equation (5) has three basic ideas embedded in its structure:

- there is an **early visual bottleneck**, in which the high-dimensional general purpose feature representation $\mathbf{\Psi}_t$ is greatly reduced in dimension (in this case, from the 4096 features of VGG's FC6 layer, to 1) prior to combination with action space,
- there is a **multiplicative interaction** between the action vector and (bottlenecked) visual features, and
- there is **symmetry**, e.g. the first term of the formula is the sign-antisymmetric partner of the second term, reflecting something about the spatial structure of the task.

In the next sections, we show these three principles can be generalized into a parameterized family of networks from which the visual bottleneck (the $W$ parameters), and decision structure (the *form* of equation (5)) can emerge naturally and efficienty via learning for any given task of interest.

### 3.1 THE EMS MODULE

In this section we define a generic ReMaP module which is lightweight, encodes all three generic design principles from the "perfect" formula, and uses only a small number of learnable parameters.

Define the *concatenated square* nonlinearity as

$$\mathbf{Sq} : x \longmapsto x \oplus x^2$$

and the *concatenated ReLU* nonlinearity (Shang et al. (2016)) as

$$\mathbf{CReLU} : x \longmapsto \mathbf{ReLU}(x) \oplus \mathbf{ReLU}(-x)$$

where $\oplus$ denotes vector concatenation. The **CReS** nonlinearity is then defined as the composition of **CReLU** and **Sq**, e.g.

$$\mathbf{CReS}(x) : x \longmapsto \mathbf{ReLU}(x) \oplus \mathbf{ReLU}(-x) \oplus \mathbf{ReLU}^2(x) \oplus \mathbf{ReLU}^2(-x).$$

The **CReS** nonlinearity introduces multiplicative interactions between its arguments via its **Sq** component and symmetry via its use of **CReLU**.

**Definition.** *The $(n_0, n_1, \ldots, n_k)$-Early Bottleneck-Multiplicative-Symmetric (EMS) module is the ReMaP module given by*

$$B = \mathbf{CReLU}(W_0\mathbf{\Psi} + b_0)$$
$$l_1 = \mathbf{CReS}(W_1(B \oplus a) + b_1)$$
$$l_i = \mathbf{CReS}(W_i l_{i-1} + b_i) \quad for \quad i > 1$$

*where $W_i$ and $b_i$ are learnable parameters, $\mathbf{\Psi}$ are features from the fixed visual encoding network, and $a$ is the action vector in $\mathcal{A}$.*

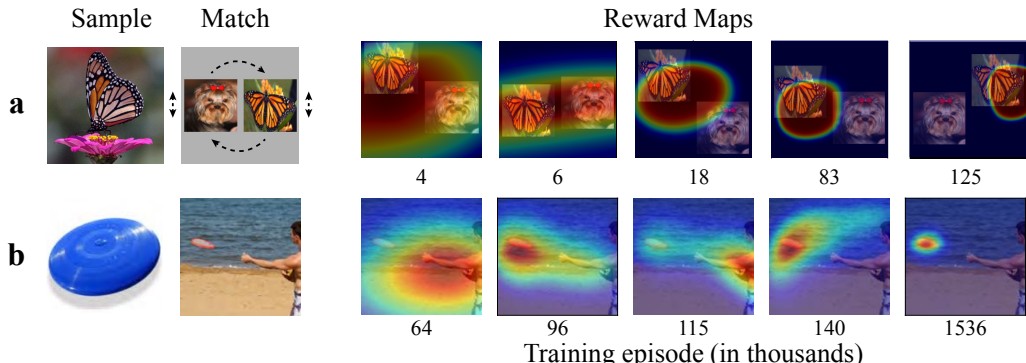

Figure 3: **Decision interfaces emerge naturally over the course of training.** The ReMaP modules allow the agent to discover the implicit interfaces for each task. We observe that learning generally first captures the emergence of natural physical constructs before learning task-specific decision rules. Examples of this include: **a.** onscreen "buttons" appearing on the match screen of an MTS task before the specific semantic meaning of each button is learned (arrows indicate random motion), and **b.** the general discovery of objects and their boundaries before the task-specific category rule is applied. This image is best viewed in color.

The EMS structure builds in each of the three principles described above. The $B$ stage represents the early bottleneck in which visual encoding inputs are bottlenecked to size $n_0$ before being combined with actions, and then performs $k$ **CReS** stages, introducing multiplicative symmetric interactions between visual features and actions. From this, the "perfect" module definition for the binary SR task in eq. (5) then becomes a special case of a two-layer EMS module. Note that the visual features to be bottlenecked can be from any encoder; in practice, we work with both fully connected and convolutional features of the VGG-16 backbone.

In the experiments that follow, we compare the EMS module to a wide variety of alternative control motifs, in which the early bottleneck, multiplicative, and symmetric features are ablated. Multiplicative nonlinearity and bottleneck ablations use a spectrum of more standard activation functions, including **ReLU**, tanh, sigmoid, elu (Clevert et al., 2015), and **CReLU** forms. In late bottleneck (fully-ablated) architectures – which are, effectively, "standard" multi-layer perceptrons (MLPs) – action vectors are concatenated directly to the output of the visual encoder before being passed through subsequent stages. In all, we test 24 distinct architectures. Detailed information on each can be found in the Supplement.

## 3.2 EXPERIMENTS

We compared each architecture across 12 variants of visual SR, MTS, and localization tasks, using fixed visual encoding features from layer FC6 of VGG-16. Task variants ranged in complexity from simple (e.g. a binary SR task with ImageNet categories) to more challenging (e.g. a many-way ImageNet MTS task with result buttons appearing in varying positions on each trial). The most complex tasks are two variants of localization, either with a single main salient object placed on a complex background (similar to images used in Yamins & DiCarlo (2016)), or complex scenes from MS-COCO (see Fig. 3b). Details of the tasks used in these experiments can be found in the Supplement. Module weights were initialized using a normal distribution with $\mu = 0.0, \sigma = 0.01$, and optimized using the ADAM algorithm (Kingma & Ba (2014)) with parameters $\beta_1 = 0.9, \beta_2 = 0.999$ and $\epsilon = 1e-8$. Learning rates were optimized on a per-task, per-architecture basis in a cross-validated fashion. For each architecture and task, we ran optimizations from five different initialization seeds to obtain mean and standard error due to initial condition variability. For fully-ablated "late-bottleneck" modules, we measured the performance of modules of three different sizes (small, medium, and large), where the smallest version is equivalent in size to the EMS module, and the medium and large versions are much larger (Table S1).

**Emergence of Decision Structures:** A key feature of ReMaP modules is that they are able to discover *de novo* the underlying output domain spaces for a variety of qualitatively distinct tasks (Fig. 3; more examples in Fig. S2). The emergent decision structures are highly interpretable and reflect the true interfaces that the environment implicitly defines. The spatiotemporal patterns of learning are robust across tasks and replicable across initial seedings, and thus might serve as a candidate model of interface use and learning in humans. In general, we observe that the modules typically discover

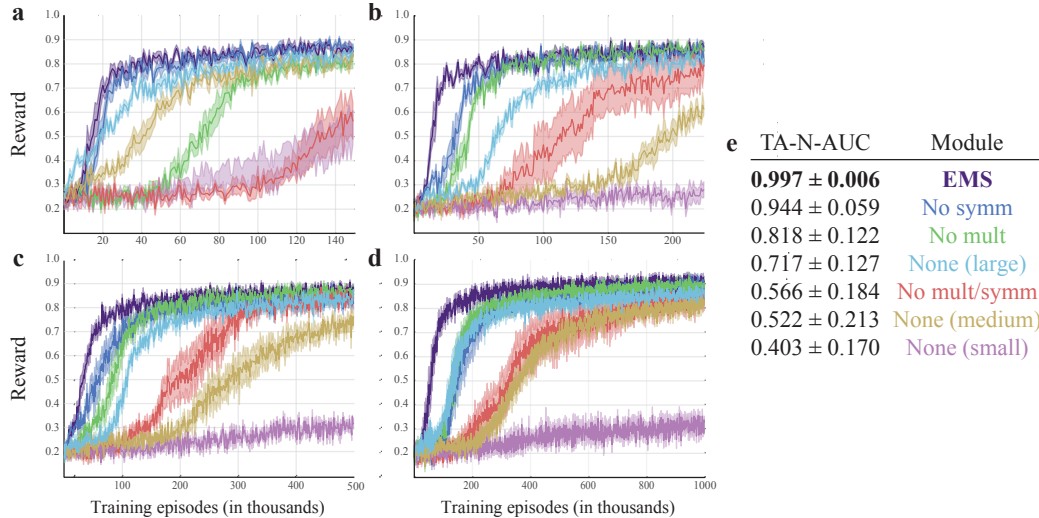

Figure 4: **EMS modules as components of an efficient visual learning system.** Validation reward obtained over the course of training for modules on **a.** 4-way stimulus-response with a reward map split into four quadrants, **b.** 2-way MTS with randomly moving match templates, **c.** 4-way MTS with two randomly moving class templates shown at a time, and **d.** 4-way MTS with four randomly positioned images shown at a time. Lines indicate mean reward over five different weight initializations. For clarity, seven of the total 24 tested architectures are displayed (see results for remaining architectures in Supplement). **e.** The TA-N-AUC metric is the area under the learning curve, normalized to the highest performing module within a task (over all 24 modules), averaged across all 12 tasks. Error values are standard deviations from the mean TA-N-AUC.

the underlying "physical structures" needed to operate the task interface before learning the specific decision rules needed to solve the task.

For example, in the case of a discrete MTS categorization task (Fig. 3a), this involves the quick discovery of onscreen "buttons" corresponding to discrete action choices before these buttons are mapped to their semantic meaning. In the case of in the MS-COCO MTS task (Fig. 3b), we observe the initial discovery of high salience object boundaries, and followed by category-specific refinement. It is important to note that the visual backbone was trained on a categorization task, quite distinct from the localization task in MS-COCO MTS. Thus, the module had to learn this very different decision structure, as well as the class boundaries of MS-COCO, from scratch during training.

**Efficiency of the EMS module:** The efficiency of learning was measured by computing the task-averaged, normalized area under the learning curve (TA-N-AUC) for each of the 24 modules tested, across all 12 task variants. Fig. 4a-d shows characteristic learning curves for several tasks, summarized in the table in Fig. 4e. Results for all architectures for all tasks are shown in Supplement Figure S1. We find that the EMS module is the most efficient across tasks (0.997 TA-N-AUC). Moreover, the EMS architecture always achieves the highest final reward level on each task.

Increasing ablations of the EMS structure lead to increasingly poor performance, both in terms of learning efficiency and final performance. Ablating the low-order polynomial interaction (replacing **Sq** with **CReLU**) had the largest negative effect on performance (0.818 TA-N-AUC), followed in importance by the symmetric structure (0.944 TA-N-AUC). Large fully-ablated models (no bottleneck, using only **ReLU** activations) performed significantly worse than the smaller EMS module and the single ablations (0.717 TA-N-AUC), but better than the module with neither symmetry nor multiplicative interactions (0.566 TA-N-AUC). Small fully-ablated modules with the same number of parameters as EMS were by far the least efficient (0.403 TA-N-AUC) and oftentimes achieved much lower final reward. In summary, the main conceptual features by which the special-case architecture in eq. (5) solves the binary SR task are both individually helpful, combine usefully, and can be parameterized and efficiently learned for a variety of visual tasks. These properties are critical to achieving effective task learning compared to standard MLP structures.

In a second experiment focusing on localization tasks, we tested an EMS module using convolutional features from the fixed VGG-16 feature encoder, reasoning that localization tasks could benefit from finer spatial feature resolution. We find that using visual features with explicit spatial information

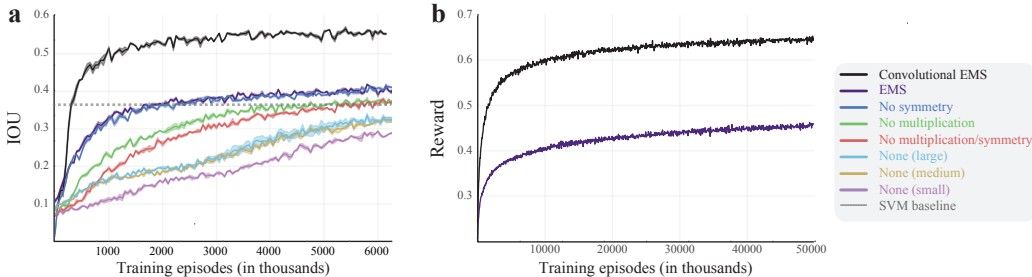

Figure 5: **Convolutional bottlenecks allow for fine resolution localization and detection in complex scenes. a.** Mean Intersection over Union (IoU) obtained on the localization task. **b.** Reward obtained on the MS-COCO match-to-sample variant. Both of these require their visual systems to accomodate for finer spatial resolution understanding of the scene, and more precise action placement than the SR or (non-COCO) MTS tasks. The convolutional variant of the EMS module uses skip connections from the conv5 and FC6 layers of VGG-16 as input, whereas the standard EMS uses only the FC6 layer as input.

substantially improves task performance and learning efficiency on these tasks (Fig. 5). To our knowledge, our results on MS-COCO are the first demonstrated use of reinforcement learning to achieve instance-level object segmentations. Reward curves (measuring bounding box IoU) in Fig. 5a show little difference between any of the late bottleneck modules at any size. The only models to consistently achieve an IoU above 0.4 are the EMS-like variants, especially with convolutional features. For context, a baseline SVR trained using supervised methods to directly regress bounding boxes using the same VGG features results in an IoU of 0.369.

## 4 DYNAMIC NEURAL VOTING FOR TASK SWITCHING

So far, we've considered the case where the TouchStream consists of only one task. However, agents in real environments are often faced with having to switch between tasks, many of which they may be encountering for the first time. Ideally, such agents would repurpose knowledge from previously learned tasks when it is relevant to a new task.

Formally, we now consider environment policies consisting of sequences of tasks $\mathcal{T} = \{\tau_1, \tau_2, ..., \tau_\Omega\}$, each of which may last for an indeterminate period of time. Consider also a set of modules $\mathcal{M}$, where each module corresponds to a task-specific policy $\pi_\omega(a \mid x) = p(a_t \mid x_t, \tau_\omega)$. When a new task begins, we cue the agent to allocate a new module $M_{\Omega+1}$ which is added to the set of modules $\mathcal{M}$. In the learning that follows allocation, the weights in old modules are held fixed while the parameters in the new module $M_{\Omega+1}$ are trained. However, the output of the system is not merely the output of the new module, but instead is a dynamically allocated mixture of pathways through the computation graphs of the old and new modules. This mixture is determined by a *meta-controller* (Fig. 6). The meta-controller is itself a neural network which learns a dynamic distribution over (parts of) modules to be used in building the composite execution graph. Intuitively, this composite graph is composed of a small number of relevant pathways that mix and match parts of existing modules to solve the new task, potentially in combination with new module components that need to be learned.

### 4.1 DYNAMIC NEURAL VOTING

We define a meta-controller that assigns weights to each layer in each module in $\mathcal{M}$. Let $p_\omega^i$ be the weight associated with the $ith$ layer in module $\omega$. These weights are probabilistic on a per layer basis, e.g. $p_\omega^i \geq 0$ and $\sum_\omega p_\omega^i = 1$ and can be interpreted as the probability of the controller selecting the $ith$ layer $l_\omega^i$ for use in the execution graph, with distribution $\pi_i = \{p_\omega^i\}$. For such an assignment of weights, the composite execution graph defined by the meta-controller is generated by computing the sum of the activations of all the components at layer $i$ weighted by the probabilities $p_\omega^i$. These values are then passed on to the next layer where this process repeats. Mathematically, the composite layer at stage $i$ can be expressed as

$$\tilde{l}_\mathcal{M}^i = \sum_\omega p_\omega^i M_\omega^i(\tilde{l}_\mathcal{M}^{i-1}) = \mathbb{E}_{\pi_i}\left[M_\omega^i(\tilde{l}_\mathcal{M}^{i-1})\right]. \tag{6}$$

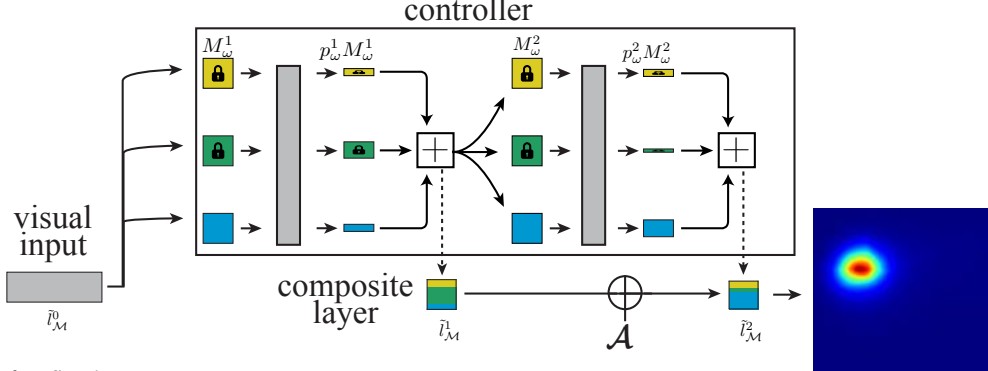

Figure 6: **The Dynamic Neural Voting Controller.** Dynamic Neural Voting solves new tasks by computing a composite execution graph through previously learned and newly allocated modules. Shown here is an agent with two existing modules (yellow and green), as one newly allocated module is being learned (blue). For each layer, the controller takes as input the activations of all three modules and outputs a set of "voting results" — probabilistic weights to be used to scale activations within the corresponding components. Voting can be done on either a per-layer basis or a per-unit basis, for clarity only the layer voting method is depicted. The weighted sum of these three scaled outputs is used as input to the next stage in the computation graph. If the new task can be solved through a combination of existing module components, these will be weighted highly, while the new module will be effectively unused e.g. is assigned low weights. If however the task is quite different than previously solved tasks, the new module will play a larger role in the execution graph as it learns to solve the task.

where $M_\omega^i(\cdot)$ is the operator that computes the $ith$ layer of module $\omega$, and $\tilde{l}^0 := \psi(x_t)$ is the original encoded input state.

The question now is, where do these probabilistic weights come from? The core of our procedure is a *dynamic neural voting* process in which the controller network learns a Boltzmann distribution over module activations to maximize reward prediction accuracy. This process is performed at each module layer, where the module weightings for a given layer are conditioned on the results of voting at the previous layer. That is,

$$\boldsymbol{p}^i = \mathbf{softmax}\left[ W^i \left( \bigoplus_\omega M_\omega^i(\tilde{l}_{\mathcal{M}}^{i-1}) \right) + b^i \right] \tag{7}$$

where $\boldsymbol{p}^i = (p_0^i, p_1^i, ..., p_\Omega^i)$ are the module weights at layer $i$, $\oplus$ is concatenation, and $W^i \in \mathbb{R}^{(\Omega \cdot L) \times \Omega}$ is a learnable weight matrix of the controller.

This voting procedure operates in an online fashion, such that the controller is continuously learning its meta-policy while the agent is taking actions. As defined, the meta-controller constitutes a fully-differentiable neural network and is learned by gradient descent online.

A useful refinement of the above mechanism involves voting across the *units* of $\mathcal{M}$. Specifically, the meta-controller now assigns probabilistic weights $p_\omega^{i,j}$ to neuron $n_\omega^{i,j}$ (the $jth$ unit in layer $i$ of module $\omega$). In contrast to the layer-voting scheme, the dynamically generated execution graph computed by the meta controller now becomes composite neurons with activations:

$$\tilde{n}_{\mathcal{M}}^{i,j} = \sum_\omega p_\omega^{i,j} M_\omega^{i,j}(\tilde{l}_{\mathcal{M}}^{i-1}) = \mathbb{E}_{\pi_i,j}\left[ M_\omega^{i,j}(\tilde{l}_{\mathcal{M}}^{i-1}) \right]. \tag{8}$$

which are concatenated to form the composite layer $\tilde{l}_{\mathcal{M}}^i$. The generalization of equation (7) to the single-unit voting scheme then becomes:

$$\boldsymbol{p}^{i,j} = \mathbf{softmax}\left[ W^{i,j} \left( \bigoplus_\omega M_\omega^{i,j}(\tilde{l}_{\mathcal{M}}^{i-1}) \right) + b^{i,j} \right] \tag{9}$$

where $\boldsymbol{p}^{i,j} = (p_0^{i,j}, p_1^{i,j}, ..., p_\Omega^{i,j})$ are the unit-level weights across modules, and $W^{i,j} \in \mathbb{R}^{\Omega \times \Omega}$.

Empirically, we find that the initialization schemes of the learnable controller parameters are an important consideration in the design, and that two specialized transformations also contribute slightly to its overall efficiency. For details on these, please refer to the Supplement.

The dynamic neural voting mechanism achieves meta-control through a neural network optimized online via gradient descent while the modules are solving tasks, rather than a genetic algorithm that operates over a longer timescale as in the work of Fernando et al. (2017). Moreover, in contrast to the work of Rusu et al. (2016) the voting mechanism eliminates the need for fully-connected adaptation layers between modules, thus substantially reducing the number of parameters required for transfer.

## 4.2 Switching Experiments

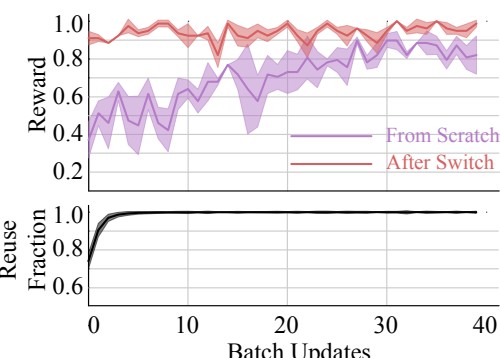

Figure 7: **Dyanmic Neural Voting quickly corrects for "no-switch" switches.** Although a new module is allocated for each task transition, if the new task is identitcal to the original task, the controller quickly learns to reuse the old module components. **Top:** post-switching learning curve for the EMS module on a binary stimulus-response task, after being trained on the same task. For clarity, only the Layer Voting method is compared against a baseline module trained from scratch. **Bottom:** fraction of the original module reused over the course of post-switch learning, calculated by averaging the voting weights of each layer in the original module.

**"No-switch" switches:** Our first experiments tested how the dynamic neural voting mechanism would respond to "no-switch" switches, i.e. ones in which although a switch cue was given and a new module allocated, the environment policy's task did not actually change (Fig 7). We find that in such cases, performance almost instantly approaches pre-switch levels (e.g. there is very little penalty in attempting an uneccessary switch). Moreover, we find that the weightings the controller applies to the new module is low: in other words, the system recognizes that no new module is needed and acts accordingly by concentrating its weights on the existing module. These results show that, while we formally assume that the agent is cued as when task switches occurs, in theory it could implement a completely autonomous monitoring policy, in which the agent simply runs the allocation procedure if a performance "anomoly" occurs (e.g. a sustained drop in reward). If the system determines that the new module was unneeded, it could simply reallocate the new module for a later task switch. In future work, we plan to implement this policy explicitly.

**"Real" switches:** We next tested how the dynamic voting controller handled switches in which the environment policy substantially changed after the switching cue. Using both the EMS module and (for control) the large fully-ablated module as described in § 3.2, the dynamic neural voting controller was evaluated on 15 switching experiments using multiple variants of SR and MTS tasks. Specifically, these 15 switches cover a variety of distinct (but not mutually exclusive) switching types including:

- addition of new classes to the dataset (switch indexes 2, 7, 11 in the table of Fig. 8)
- replacing the current class set entirely with a new non-overlapping class set (switch ids. 1, 3)
- addition of visual variability to a previously less variable task (switch id. 6)
- addition of visual interface elements e.g. new buttons (switch id. 8)
- transformation of interface elements e.g. screen rotation (switch ids. 12, 13, 14, 15)
- transitions between different task paradigms e.g. SR to MTS tasks and vice-versa (switch ids. 4, 5, 9, 10).

Controller hyperparameters were optimized in a cross-validated fashion (see Appendix G.1), and optimizations for three different initialization seeds were run to obtain mean and standard error.

Figures 8a and b show characteristic post-switch learning curves for the EMS module for both the Layer Voting and Single-Unit Voting methods. Additional switching curves can be found in the Supplement. Cumulative reward gains relative to learning from scratch were quantified by Relative Gain in AUC: $RGain = \frac{AUC(M^{switch}) - AUC(M)}{AUC(M)}$, where $M$ is the module trained from scratch on

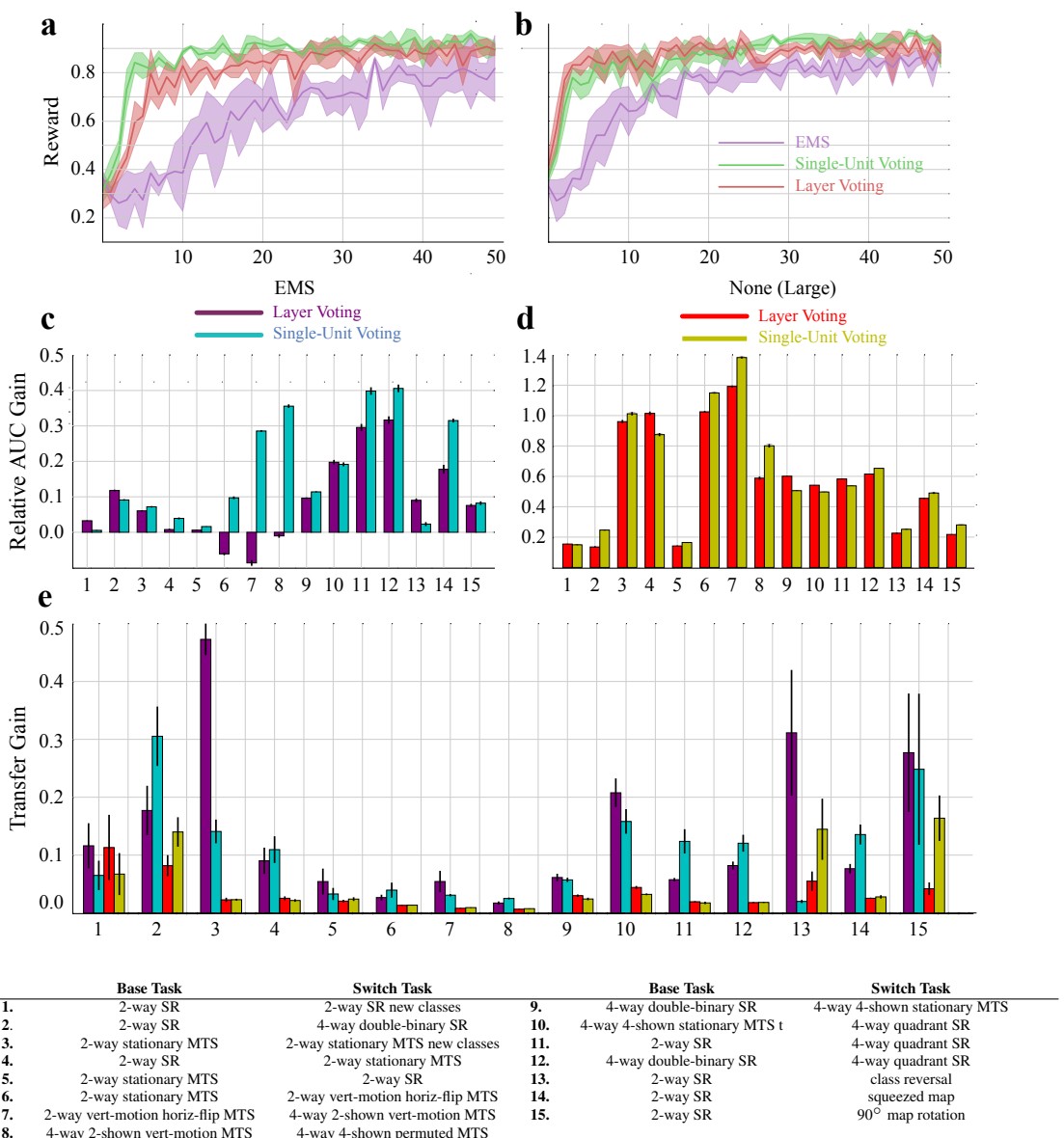

| | **Base Task** | **Switch Task** | | **Base Task** | **Switch Task** |
|---|---|---|---|---|---|
| **1.** | 2-way SR | 2-way SR new classes | **9.** | 4-way double-binary SR | 4-way 4-shown stationary MTS |
| **2.** | 2-way SR | 4-way double-binary SR | **10.** | 4-way 4-shown stationary MTS t | 4-way quadrant SR |
| **3.** | 2-way stationary MTS | 2-way stationary MTS new classes | **11.** | 2-way SR | 4-way quadrant SR |
| **4.** | 2-way SR | 2-way stationary MTS | **12.** | 4-way double-binary SR | 4-way quadrant SR |
| **5.** | 2-way stationary MTS | 2-way SR | **13.** | 2-way SR | class reversal |
| **6.** | 2-way stationary MTS | 2-way vert-motion horiz-flip MTS | **14.** | 2-way SR | squeezed map |
| **7.** | 2-way vert-motion horiz-flip MTS | 4-way 2-shown vert-motion MTS | **15.** | 2-way SR | 90° map rotation |
| **8.** | 4-way 2-shown vert-motion MTS | 4-way 4-shown permuted MTS | | | |

**Task switching with Dynamic Neural Voting.** Post-Switching learning curves for the EMS module on the 4-way Quadrant SR task after learning **a.** 2-way SR task and **b.** a 4-way MTS task with 4 match screen class templates. Both the Layer Voting method and Single-Unit Voting method are compared against a baseline module trained on the second task from scratch. Across all twelve task switches, we evaluate the Relative Gain in AUC over baseline (RGain) using both voting methods for **c.** the EMS module and **d.** the large-sized fully-ablated late bottleneck MLP. **e.** Transfer Gain (TGain) metrics are compared for both module types for each of the voting mechanisms. Colors are as in c. (EMS module) and d. (fully-ablated module).

Figure 8

the second task, and $M^{switch}$ is the module transferred from an initial task using the dynamic voting controller. We find that the dynamic voting controller allows for rapid positive transfer of both module types across all 15 task switches, and the general Single-Unit voting method is a somewhat better transfer mechanism than the Layer Voting method (Fig. 8c). Both the EMS module and the large fully-ablated module, which was shown to be inefficient on single-task performance in § 3.2, benefit from dynamic neural voting (Fig. 8 d).

**EMS modules are more "switchable":** To quantify how *fast* switching gains are realized, we use Transfer Gain: $TGain = \frac{\Delta_{max}}{T_{\Delta_{max}}}$, where $T_{\Delta_{max}} = \text{argmax}(\Delta_t)$ is the time where the maximum amount of reward difference between $M^{switch}$ and $M$ occurs, and $\Delta_{max}$ is the reward difference at that time. Qualitatively, a high score on the Transfer Gain metric indicates that a large amount of relative reward improvement has been achieved in a short amount of time (see Figure S7 for a graphical illustration of the relationship between the $RGain$ and $TGain$ metrics). While both the EMS and large fully-ablated modules have positive Transfer Gain, EMS scores significantly higher on this metric, i.e. is significantly more "switchable" than the large fully-ablated module (Fig. 8e). We hypothesize that this is due to the EMS module being able to achieve high task performance with significantly fewer units than the larger fully-ablated module, making the former easier for the dynamic neural voting controller to operate on.

## 5 CONCLUSION AND FUTURE DIRECTIONS

In this work, we introduce the TouchStream environment, a continual reinforcement learning framework that unifies a wide variety of spatial decision-making tasks within a single context. We describe a general algorithm (ReMaP) for learning light-weight neural modules that discover implicit task interfaces within this large-action/state-space environment. We show that a particular module architecture (EMS) is able to remain compact while retaining high task performance, and thus is especially suitable for flexible task learning and switching. We also describe a simple but general dynamic task-switching architecture that shows substantial ability to transfer knowledge when modules for new tasks are learned.

A crucial future direction will be to expand insights from the current work into a more complete continual-learning agent. We will need to show that our approach scales to handle dozens or hundreds of task switches in sequence. We will also need to address issues of how the agent determines *when* to build a new module and how to *consolidate* modules when appropriate (e.g. when a series of tasks previously understood as separate can be solved by a single smaller structure). It will also be critical to extend our approach to handle visual tasks with longer horizons, such as navigation or game play with extended strategic planning, which will likely require the use of recurrent memory stores as part of the feature encoder.

From an application point of view, we are particularly interested in using techniques like those described here to produce agents that can autonomously discover and operate the interfaces present in many important real-world two-dimensional problem domains, such as on smartphones or the internet (Grossman, 2007). We also expect many of the same spatially-informed techniques that enable our ReMaP/EMS modules to perform well in the 2-D TouchStream environment will also transfer naturally to a three-dimensional context, where autonomous robotics applications (Devin et al., 2016) are very compelling.

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

## SUPPLEMENTARY MATERIAL

## A  TASK VARIANTS

The EMS module and all ablation controls were evaluated on a suite of 13 stimulus-response, match-to-sample, localization, and MS-COCO MTS variants:

1. 2-way SR - standard binary SR task
2. 4-way double binary SR - four class variant of SR, where each class is assigned either to the right or left half of the action space
3. 4-way quadrant SR - four class variant of SR, where each class is assigned to only a quadrant of the action space
4. 2-way stationary MTS - standard binary MTS task with stereotyped and non-moving match screens

5. 2-way stationary horiz-flip MTS - two class variant MTS task where the match templates' horizontal placement is randomly chosen, but confined within the same vertical plane

6. 2-way stationary vert-motion MTS - two class variant MTS task where the match templates' vertical position is randomly chosen, but each class is confined to a specific side

7. 2-way stationary vert-motion horiz-flip MTS - two class variant MTS task where the match templates' positions are completely random

8. 4-way 2-shown MTS - four class variant MTS task where only two class templates are shown on the match screen (appearing with random horizontal location as well)

9. 4-way 2-shown vert-motion MTS - same as above, but with random vertical motion for the templates

10. 4-way 4-shown stationary MTS - four class variant MTS task where all four class templates are shown on the match screen, but with fixed positions.

11. 4-way 4-shown permuted MTS - same as above, but with randomly permuted locations of all match templates

12. Localization - Localization task

13. MS-COCO MTS - 80-way MTS task using the MS-COCO detection challenge dataset, where match screens are randomly samples scenes from the dataset

## B  EXPERIMENT DETAILS AND DATASETS

**Stimulus-Response Experiment Details:** Image categories used are drawn from the Image-Net 2012 ILSVR classification challenge dataset Deng et al. (2009). Four unique object classes are taken from the dataset: Boston Terrier, Monarch Butterfly, Race Car, and Panda Bear. Each class has 1300 unique training instances, and 50 unique validation instances.

**Match-To-Sample Experiment Details:** Sample screen images drawn from the same Image-Net class set as the Stimulus-Response tasks. One face-centered, unobstructed class instance is also drawn from the Image-Net classification challenge set and used as a match screen template image for that class. Class template images for the match screen were held fixed at 100x100 pixels. For all variants of the MTS task, we keep a six pixel buffer between the edges of the screen and the match images, and a twelve pixel buffer between the adjacent edges of the match images themselves. Variants without vertical motion have the match images vertically centered on the screen.

**Localization Experiment Details:** The Localization task uses synthetic images containing a single main salient object placed on a complex background (similar to images used in Yamins & DiCarlo (2016); Yamins et al. (2014)). There are a total of 59 unique classes in this dataset. In contrast to other single-class localization datasets (e.g. Image-Net) which are designed to have one large, face-centered, and centrally-focused object instance and for which a trivial policy of "always poke in image corners" could be learned, this synthetic image set offers larger variance in instance scale, position, and rotation so the agent is forced into learning non-trivial policies requiring larger precision in action selection.

**MS-COCO MTS Experiment Details** This task uses the entire MS-COCO detection challenge dataset Lin et al. (2014). On every timestep, a sample screen chosen from one of the 80 MS-COCO classes. These are constructed to be large, unobstructed, face centered representations of the class. For the match screen, we sample a random scene from MS-COCO containing any number of objects, but containing at least a single instance of the sample class. The agent is rewarded if its action is located inside any instance of the correct class. Both modules use sample actions from a low-temperature Boltzmann policy from eq. (4), which was empirically found to result in more precise reward map prediction.

## C  MODULES

### C.1  UNITS PER LAYER

Table S1 aggregates the number of units per layer for the EMS and ablated modules which was used when conducting single-task and task-switching experiments. Only fully-connected modules' layer sizes are shown here. For details on the convolutional bottleneck EMS module, please refer to C.2.

Table S1: Number of units per layer for investigated modules

| Base-task | EMS | No symm | No Mult | No mult/symm | None-Small | None-Med | None-Large |
|---|---|---|---|---|---|---|---|
| SR | 8 | 8 | 8 | 8 | 8 | 128 | 512 |
| MTS | 32 | 32 | 32 | 32 | 32 | 128 | 512 |
| LOC | 128 | 128 | 128 | 128 | 128 | 512 | 1024 |

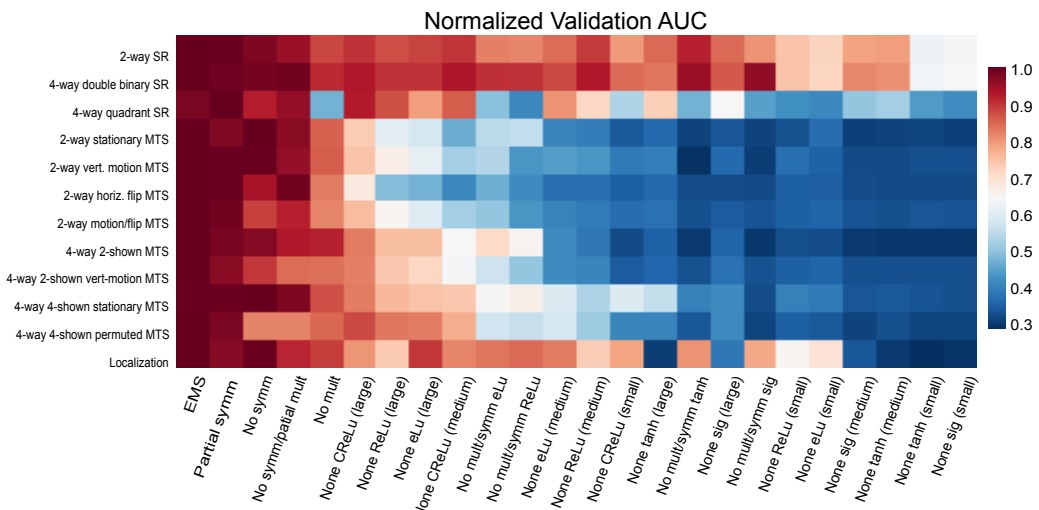

Figure S1: **Exhaustive module performance study** of the EMS module and 23 ablation control modules, measured as the Area Under the Curve for all SR, MTS, and LOC task variants. Shown is the AUC normalized to the highest performing module in a task. Results in fig. 4 have further averaged this over the vertical task axis, and report only a salient subset of the ablations.

## C.2 The Convolutional-EMS Module

This is a "Convolutional Bottleneck" extension of the EMS module shown in the paper, where skip connections link the conv5 and the FC6 representation of the visual backbone. Here, the "scene-level" representation stored in the FC6 ReMaP memory buffer is tiled spatially to match the present convolution dimensions (here 14x14), and concatenated onto its channel dimension. A series of 1x1 convolutions plays the role of a shallow visual bottleneck, before the activations are vectorized and concatenated with $\mathcal{A}$ as input to the **CReS** layers of the standard EMS module.

The results in the paper are shown for a bottleneck consisting of a single tanh and two **CReS** convolutions, with 128 units each. The Downstream layers use 128 units each as well.

The motivation for the convolutional bottleneck is that lower-level features are useful for complex spatial tasks such as Localization and Object Detection, and hence may result in a more precise policy. By tiling the entire scene-level representation along the convolution layer's channel dimension, a form of multiplicative template-matching is possible between objects that must be memorized (e.g. MS-COCO MTS templates) and what is inside the present scene.

## D Exhaustive Ablation Study

In all, we investigated 23 distinct ablations on the EMS module, across all twelve task variants outlined in sec A (Fig. S1). Symmetry ablations replace **CReS** with the activation $x \mapsto \mathbf{ReLU}(x) \oplus x^2$ Multiplicative ablations are denoted by specifying the nonlinearity used in place of **CReS** (where this is one of **ReLU**, tanh, sigmoid, elu Clevert et al. (2015), or **CReLU** Shang et al. (2016)). This additionally includes one partial symmetry ablation (denoted "partial symm") where only the visual bottleneck is symmetric, and one which ablates the **ReLU** from the "no symm" module (denoted "no symm/partial-mult").

Table S2: Module learning rates

| | 2-way SR | 4-way double binary SR | 4-way stationary SR | 2-way stationary MTS | 2-way vert-motion MTS | 2-way horiz flip MTS | 2-way motion/flip MTS | 4-way 2-shown MTS | 4-way 2-shown vert-motion MTS | 4-way 4-shown stationary MTS | 4-way 4-shown permuted MTS | LOC |
|---|---|---|---|---|---|---|---|---|---|---|---|---|
| EMS | $10^{-3}$ | $10^{-3}$ | $10^{-3}$ | $5\cdot10^{-4}$ | $5\cdot10^{-4}$ | $5\cdot10^{-4}$ | $5\cdot10^{-4}$ | $5\cdot10^{-4}$ | $5\cdot10^{-4}$ | $5\cdot10^{-4}$ | $5\cdot10^{-4}$ | $10^{-4}$ |
| Partial symm | $10^{-3}$ | $10^{-3}$ | $10^{-3}$ | $5\cdot10^{-4}$ | $5\cdot10^{-4}$ | $5\cdot10^{-4}$ | $5\cdot10^{-4}$ | $5\cdot10^{-4}$ | $5\cdot10^{-4}$ | $5\cdot10^{-4}$ | $5\cdot10^{-4}$ | $10^{-4}$ |
| No symm | $10^{-3}$ | $10^{-3}$ | $10^{-3}$ | $10^{-3}$ | $10^{-3}$ | $10^{-3}$ | $10^{-3}$ | $10^{-3}$ | $10^{-3}$ | $10^{-3}$ | $2\cdot10^{-4}$ | $10^{-4}$ |
| No symm/partial mult | $10^{-3}$ | $10^{-3}$ | $10^{-3}$ | $10^{-3}$ | $10^{-3}$ | $10^{-3}$ | $10^{-3}$ | $10^{-3}$ | $10^{-3}$ | $10^{-3}$ | $2\cdot10^{-4}$ | $10^{-4}$ |
| No mult/symm ReLU | $10^{-3}$ | $10^{-3}$ | $10^{-3}$ | $10^{-3}$ | $10^{-3}$ | $10^{-3}$ | $10^{-3}$ | $10^{-3}$ | $10^{-3}$ | $10^{-3}$ | $10^{-3}$ | $10^{-4}$ |
| No mult/symm tanh | $10^{-3}$ | $10^{-3}$ | $10^{-3}$ | $10^{-3}$ | $10^{-4}$ | $10^{-3}$ | $10^{-3}$ | $10^{-3}$ | $10^{-3}$ | $10^{-4}$ | $10^{-4}$ | $10^{-4}$ |
| No mult/symm sig | $10^{-3}$ | $10^{-3}$ | $10^{-3}$ | $10^{-3}$ | $10^{-4}$ | $10^{-3}$ | $10^{-3}$ | $10^{-3}$ | $10^{-3}$ | $10^{-4}$ | $10^{-4}$ | $10^{-4}$ |
| No mult/symm eLU | $10^{-3}$ | $10^{-3}$ | $10^{-4}$ | $10^{-3}$ | $10^{-3}$ | $10^{-3}$ | $10^{-3}$ | $10^{-3}$ | $10^{-3}$ | $10^{-3}$ | $10^{-3}$ | $10^{-4}$ |
| No mult/symm CReLU | $10^{-3}$ | $10^{-3}$ | $10^{-3}$ | $10^{-3}$ | $10^{-3}$ | $10^{-3}$ | $10^{-3}$ | $10^{-3}$ | $10^{-3}$ | $10^{-3}$ | $10^{-3}$ | $10^{-4}$ |
| None ReLU(small) | $10^{-3}$ | $10^{-3}$ | $10^{-3}$ | $10^{-3}$ | $10^{-3}$ | $10^{-3}$ | $10^{-3}$ | $10^{-3}$ | $10^{-3}$ | $10^{-3}$ | $10^{-3}$ | $10^{-4}$ |
| None ReLU(medium) | $10^{-3}$ | $10^{-3}$ | $10^{-3}$ | $10^{-4}$ | $10^{-4}$ | $10^{-4}$ | $10^{-4}$ | $10^{-4}$ | $10^{-4}$ | $10^{-4}$ | $10^{-4}$ | $10^{-4}$ |
| None ReLU(large) | $10^{-3}$ | $10^{-3}$ | $10^{-4}$ | $10^{-4}$ | $10^{-4}$ | $10^{-4}$ | $10^{-4}$ | $10^{-4}$ | $10^{-4}$ | $10^{-4}$ | $10^{-4}$ | $10^{-4}$ |
| None tanh(small) | $10^{-4}$ | $10^{-4}$ | $10^{-4}$ | $10^{-3}$ | $10^{-3}$ | $10^{-3}$ | $10^{-3}$ | $10^{-3}$ | $10^{-4}$ | $10^{-4}$ | $10^{-4}$ | $10^{-4}$ |
| None tanh(medium) | $10^{-4}$ | $10^{-4}$ | $10^{-4}$ | $10^{-3}$ | $10^{-3}$ | $10^{-3}$ | $10^{-3}$ | $10^{-3}$ | $10^{-3}$ | $10^{-4}$ | $10^{-4}$ | $10^{-4}$ |
| None tanh(large) | $10^{-4}$ | $10^{-4}$ | $10^{-4}$ | $10^{-4}$ | $10^{-4}$ | $10^{-4}$ | $10^{-4}$ | $10^{-4}$ | $10^{-4}$ | $10^{-4}$ | $10^{-4}$ | $10^{-4}$ |
| None sig(small) | $10^{-4}$ | $10^{-4}$ | $10^{-4}$ | $10^{-3}$ | $10^{-3}$ | $10^{-3}$ | $10^{-3}$ | $10^{-3}$ | $10^{-3}$ | $10^{-4}$ | $10^{-3}$ | $10^{-4}$ |
| None sig(medium) | $10^{-4}$ | $10^{-4}$ | $10^{-4}$ | $10^{-3}$ | $10^{-3}$ | $10^{-3}$ | $10^{-3}$ | $10^{-3}$ | $10^{-4}$ | $10^{-4}$ | $10^{-4}$ | $10^{-4}$ |
| None sig(large) | $10^{-4}$ | $10^{-4}$ | $10^{-4}$ | $10^{-4}$ | $10^{-4}$ | $10^{-4}$ | $10^{-4}$ | $10^{-4}$ | $10^{-4}$ | $10^{-4}$ | $10^{-4}$ | $10^{-4}$ |
| None eLU(small) | $10^{-3}$ | $10^{-3}$ | $10^{-3}$ | $10^{-3}$ | $10^{-3}$ | $10^{-3}$ | $10^{-3}$ | $10^{-3}$ | $10^{-3}$ | $10^{-3}$ | $10^{-3}$ | $10^{-4}$ |
| None eLU(medium) | $10^{-3}$ | $10^{-3}$ | $10^{-3}$ | $10^{-4}$ | $10^{-4}$ | $10^{-4}$ | $10^{-4}$ | $10^{-4}$ | $10^{-4}$ | $10^{-3}$ | $10^{-4}$ | $10^{-4}$ |
| None eLU(large) | $10^{-4}$ | $10^{-4}$ | $10^{-3}$ | $10^{-3}$ | $10^{-4}$ | $10^{-4}$ | $10^{-4}$ | $10^{-4}$ | $10^{-4}$ | $10^{-4}$ | $10^{-4}$ | $10^{-4}$ |
| None CReLU(small) | $10^{-3}$ | $10^{-3}$ | $10^{-3}$ | $10^{-3}$ | $10^{-3}$ | $10^{-4}$ | $10^{-3}$ | $10^{-3}$ | $10^{-3}$ | $10^{-3}$ | $10^{-3}$ | $10^{-4}$ |
| None CReLU(medium) | $10^{-3}$ | $10^{-3}$ | $10^{-3}$ | $10^{-4}$ | $10^{-4}$ | $10^{-4}$ | $10^{-4}$ | $10^{-4}$ | $10^{-4}$ | $10^{-3}$ | $10^{-4}$ | $10^{-4}$ |
| None CReLU(large) | $10^{-4}$ | $10^{-4}$ | $10^{-4}$ | $10^{-4}$ | $10^{-4}$ | $10^{-4}$ | $10^{-4}$ | $10^{-4}$ | $10^{-4}$ | $10^{-4}$ | $10^{-4}$ | $10^{-4}$ |

### D.1 HYPERPARAMETERS

Learning rates for the ADAM optimizer were chosen on a per-task basis through cross-validation on a grid between $[10^{-4}, 10^{-3}]$ for each architecture. Values used in the present study may be seen in Table S2.

## E ADDITIONAL MS-COCO REWARD MAPS

Five additional reward map examples for the MS-COCO MTS task are provided in in Figure S2. Examples are plotted over the course of learning.

## F ADDITIONAL LEARNING CURVES

### F.1 SINGLE-TASK ABLATION EXPERIMENTS

Learning trajectories for seven additional tasks are provided in Figure S3. Modules capable of convergence on a task were run until this was acheived, but AUC values for a given task are calculated at the point in time when the majority of models converge.

### F.2 DYNAMIC VOTING CONTROLLER AND EMS MODULE TASK-SWITCHING EXPERIMENTS

Additional trajectories for ten unshown switching curves are provided in Figure S4.

## G DYNAMIC VOTING CONTROLLER AUGMENTATIONS

### G.1 LEARNABLE PARAMETER INITIALIZATIONS

Here we describe the weight initialization scheme that was found to be optimal for use with the dynamic voting controller. For simplicity, consider the layer-voting mechanism, with learnable

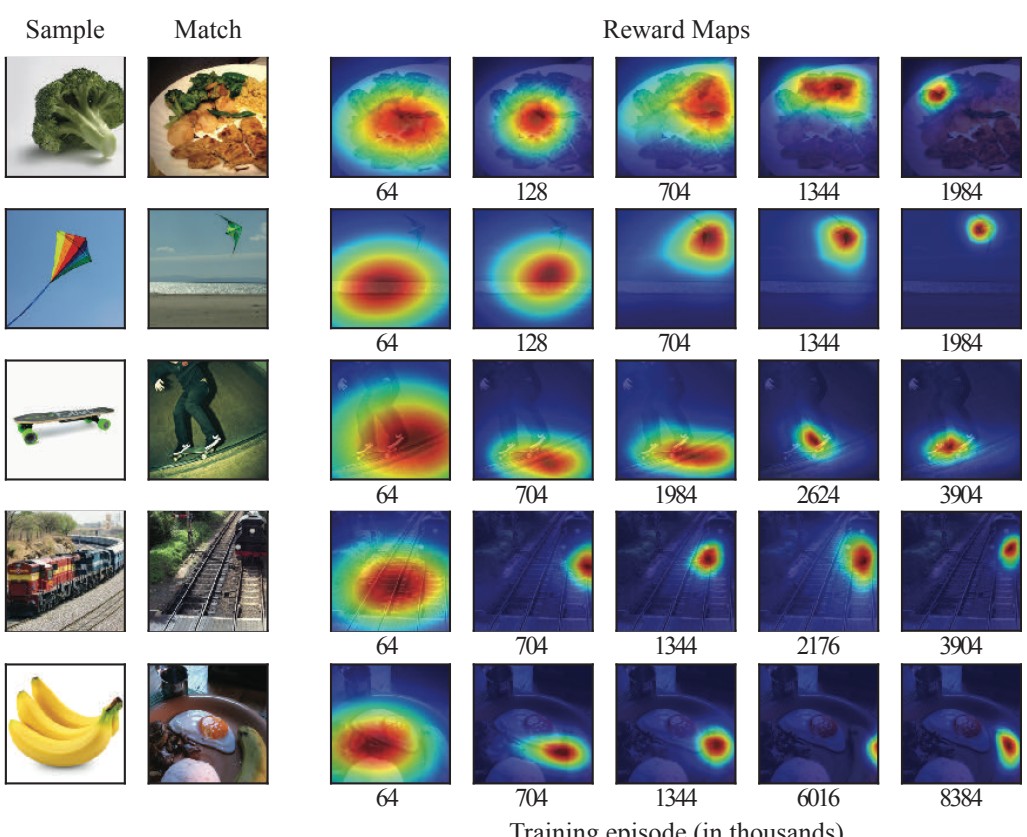

Figure S2: **Examples of the emergence of decision interfaces in MSCOCO MTS** Reward map predictions over the course of training for 5 different object classes.

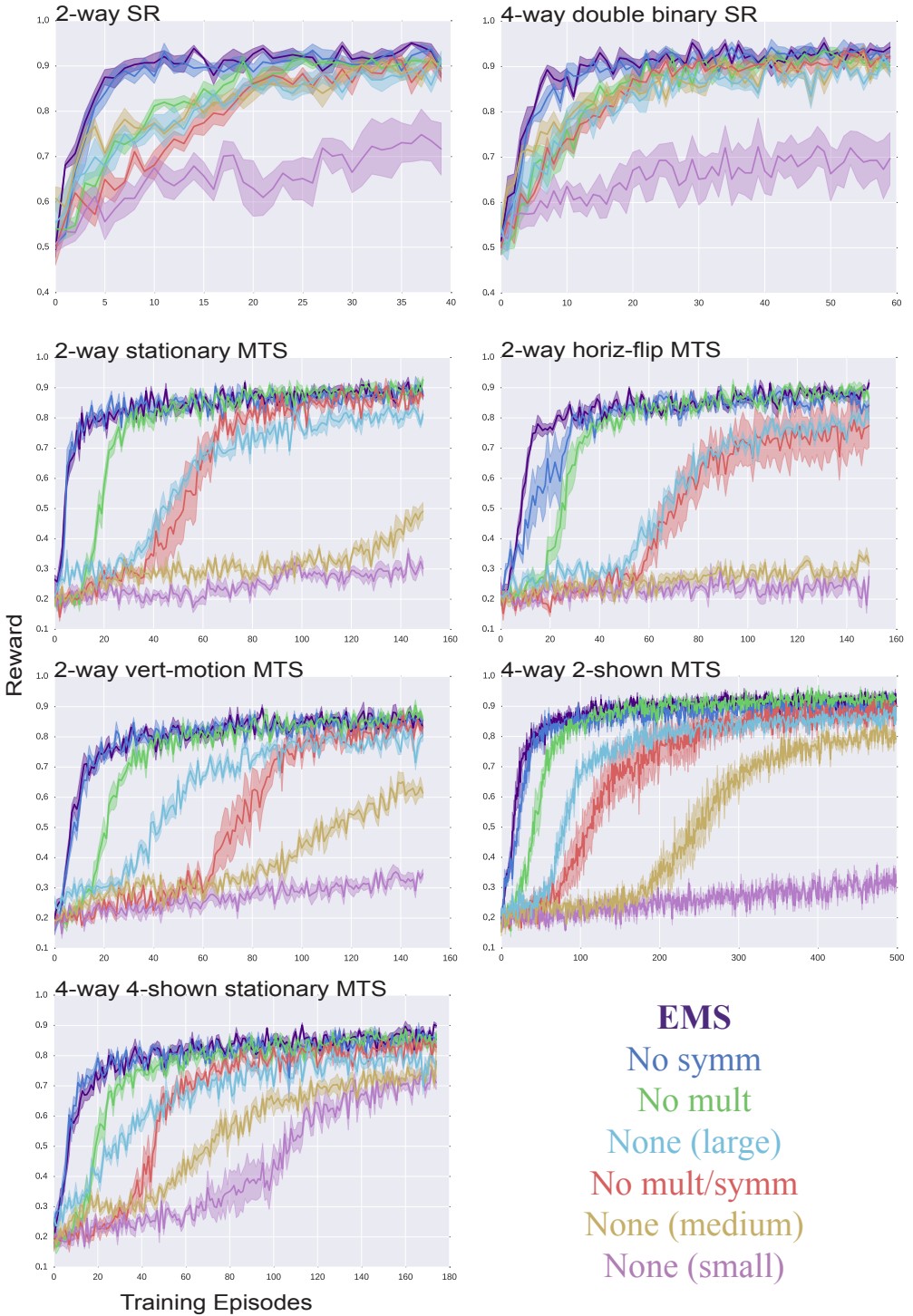

Figure S3: **Additional Single-task performance ablation Learning curves.** Seven learning curves shown for task variants not seen in the main text body. Shown are the same ablations as the main text.

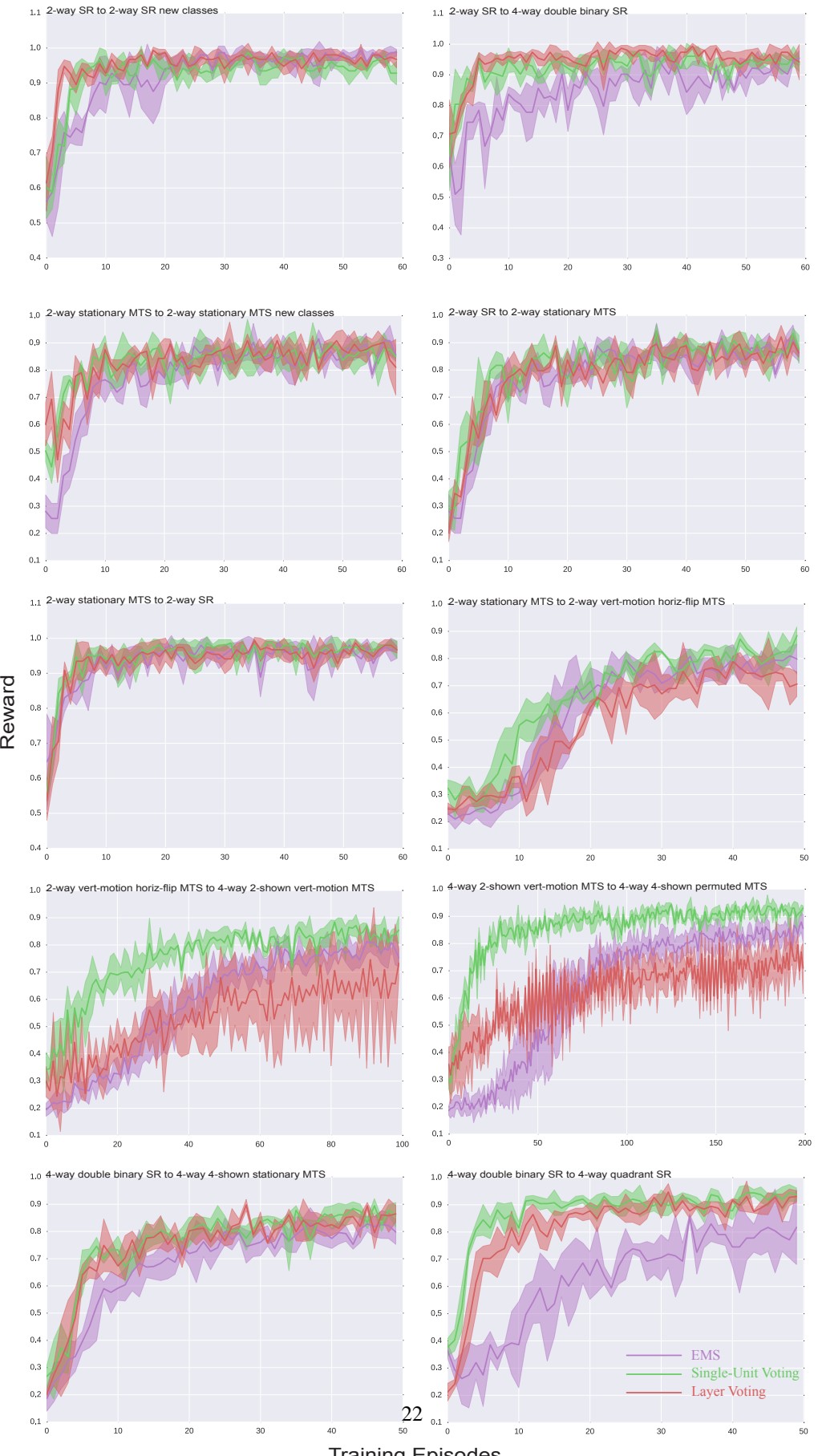

Training Episodes

weight matricies $W^i$ and biases $b^i$. The intended biasing scheme is achieved through initializing the elements of these parameters to:

$$W_\omega^i \sim \begin{cases} |\mathcal{N}(\mu^0, 0.001)| & \text{if } i = 1, \ \omega < \Omega \\ |\mathcal{N}(\mu^1, 0.001)| & \text{if } i > 1, \ \omega < \Omega \\ |\mathcal{N}(0.01, 0.001)| & \text{if } \omega = \Omega \end{cases} \tag{10}$$

$$b_\omega^i = \begin{cases} b^0 & \text{if } i = 1, \ \omega < \Omega \\ b^1 & \text{if } i > 1, \ \omega < \Omega \\ 0.1 & \text{if } \omega = \Omega \end{cases} \tag{11}$$

This initialization technique was also generalized for use with the single-unit voting mechanism.

For the switching experiments presented in section § 4.2, we sweep the hyperparameters on a narrow band around the default scheme. The ranges for these are: $\mu^0 \in [0.01, 0.005]$, $b^0 \in [0.1, 0.01]$, $\mu^1 \in [0.01, 0.02]$, and $b^1 \in [0.1, 0.2, 0.5, 1.0]$.

### G.2    TARGETED TRANSFORMATIONS

Two additional switching mechanisms were added to the controller to augment its ability to switch between taks which are remappings of the action space or reward policy of a preexisting module.

#### G.2.1    ACTION TRANSFORMATIONS

we note that efficient modules are those which can effectively produce a minimal representation of the interaction between action space $\mathcal{A}$ and observation $x_t$. If the agent's optimal action space shifts to $\mathcal{A}'$ while the remainder of the task context remains fixed, the controller should allow for rapid targeted remapping $\mathcal{A} \mapsto \mathcal{A}'$. Since we formulate the modules as ReMaP Networks, and $\mathcal{A}$ is an input feature basis, we can achieve remappings of this form through a fully-connected transformation:

$$\mathbf{a}_\tau' = f(W_a \mathbf{a}_\tau + b) \tag{12}$$

where $\mathbf{a}_\tau = [\mathbf{h}_{t-kb:t-1}, a_t]$ is the vector of action histories, and $W_a$ and $b$ embed $\mathbf{a}_\tau$ into new action space $\mathcal{A}'$ using only a small number of learnable parameters.

**Pseudo Identity-Preserving Transformation** In practice, we initialize the parameters in eq. (12) such that the transformation is *pseudo identity-preseving*, meaning that the representation learned at this level in the original module is not destroyed prior to transfer.

This is done by initializing $W_a$ to be an identity matrix $I_{|\mathbf{a}_\tau|}$ with a small amount of Gaussian noise $\epsilon \sim \mathcal{N}(0.0, \sigma^2)$ added to break symmetry. $b$ is initialized to be a vector of ones of size $|\mathbf{a}_\tau|$.

#### G.2.2    REWARD MAP TRANSFORMATIONS

Each of the $k_f$ maps $m_t(x)$ reflects the agent's uncertainty in the environment's reward policy. If the task context remains stationary, but the environment transitions to new reward schedule $\mathcal{R}'$ that no longer aligns with the module's policy $\pi$, the controller could to this transition by e.g. containing a mechanism allowing for targeted transformation of $m(x)$ and hence also $\pi$.

One complication that arises under ReMaP is that since each task-module *learns* its optimal action space internally, $m(x)$ are in the basis of $\mathcal{R}$ rather than $\mathcal{A}$. Therefore, transformations on the map distribution must also re-encode $\mathcal{A}$ before mapping to $\mathcal{R}'$.

In this work, we investigate a shallow "adapter" neural network that lives on top of the existing module and maps $\mathcal{R} \mapsto \mathcal{R}'$. Its first and second layers are defined by

$$l_1(x) = f(W_1[m(x) \odot g(\mathbf{a}_\tau), \mathbf{a}_\tau] + b_1 \tag{13}$$

$$m(x)' \propto W_2 l_1 + b_2 \tag{14}$$

where $g(\mathbf{a}_\tau)$ is a similar transformation on $\mathcal{A}$ as above, $\odot$ denotes elementwise multiplication, $W_1$ is a learnable matrix embedding into a hidden state, and $W_2 \in \mathbb{R}^{|l_1| \times |\mathcal{R}'|}$ is a learnable matrix embedding into $\mathcal{R}'$

**Pseudo Identity-Preserving Transformation** Similar to the transformation on the action space, we modify the reward-map transformation to be pseudo identity-preserving as well. This is done by modifying eq. (13) such that the original maps are concatenated on to the beginning of the transformation input vector:

$$l_1(x) = f(W_1[m(x), m(x) \odot g(\mathbf{a}_\tau), \mathbf{a}_\tau] + b_1 \tag{15}$$

The intended map-preserving transformation is accomplished via initializing $W_1$ and $W_2$ as:

$$W^{(i,j)} \sim \begin{cases} 1.0 + \mathcal{N}(0.0, \epsilon) & \text{if } i = j, \ i < \mathcal{R} \\ \mathcal{N}(0.0, \epsilon) & otherwise \end{cases} \tag{16}$$

### G.3 TARGETED TRANSFORMATION HYPERPARAMETERS

Both of the targeted transformations have several hyperparameters. We conducted a grid search to optimize these in a cross-validated fashion, on a set of test task switches designed to be solved by one of the targeted transformations (Fig. S5). Each was conducted independently of the dynamic voting controller, and independently of the other transformation. Optimal hyperparameters found in these experiments were fixed for use in the integrated dynamic voting controller, and were not further optimized afterwards.

**Action Transformation Hyperparameters** We conducted three tests using the stimulus-response paradigm: class reversal (in which the left class becomes the right class and vice-versa), a horizontal rotation of the reward boundaries (such that right becomes up and left becomes down), and a "switch" to the original task (intended to test the identity-preserving component).

In this work, we find that a single, non-activated linear transformation ($f$ in (12)) is optimal for this new state-space embedding, using $k_b * 2$ units, and initialized such that the idendity-preserving transformation weights have $\sigma = 0.01$. The learning rate for this transformation was found to be optimal at $0.1$.

**Reward Map Transformation Hyperparameters** We conducted two tests using the stimulus-response paradigm: a "squeezing" task (where there is no longer any reward dispensed on the lower half of the screen), and a "switch" to the original task (intended to test the identity-preserving component).

In this work, we find the optimal activations in eq. (15) to be $f(\cdot) = \mathbf{CReS}$ and $g(\cdot) = \mathbf{ReLU}$, with 4 units in the hidden layer. $\epsilon$ in the weight initialization scheme was found optimal at $0.001$, and an initial bias of $0.01$. The optimal learning rate for this transformation was found to be $0.01$.

### G.4 TRANSFORM ABLATION

A study was conducted to determine the relative benefit of the targeted transformations (Fig. S6), where it was determined that the primary contribution of the dynamic neural controller was in fact the voting mechanism (although the transformations did supplement this as well).

### G.5 DEPLOYMENT SCHEME OF TASK MODULES

When cued into task transition, the controller freezes the learnable parameters of the old task-module, and deploys a new unitialized task-module. The controller then initializes the action and reward map transformation networks as described in G.2 on top of the old module. These transformations are also voted on inside the dynamic neural controller at every timestep.

## H SWITCHING METRICS

Figure S7 graphically illustrates the metrics used inside the paper to quantify switching performance: $RGain$ and $TGain$.

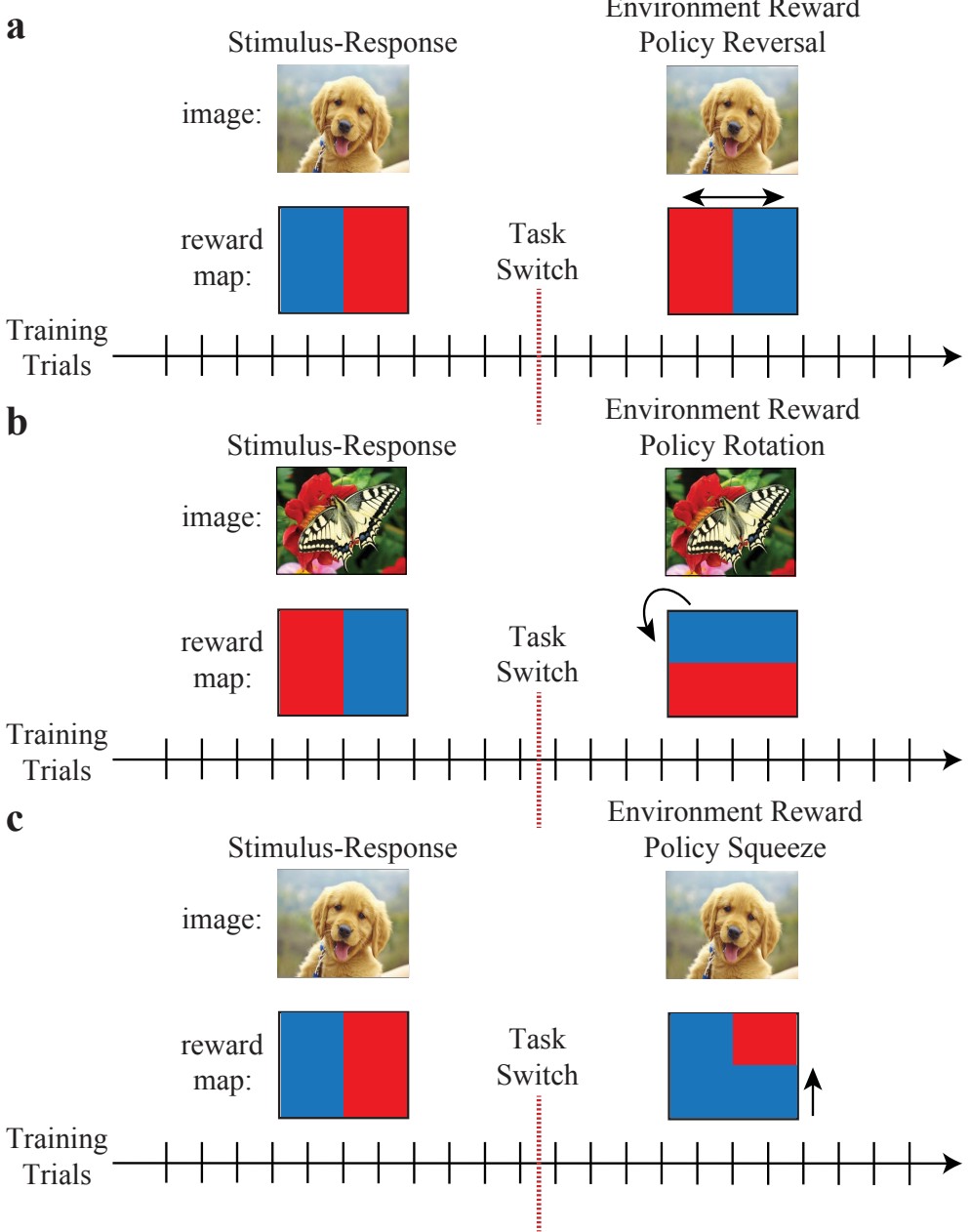

Figure S5: **Action and reward map transformation switch examples.** Three task switching experiments were performed to optimize the hyperparameters of the targeted transformations that augment the dynamic neural voting controller. These switches are also retested in the fully-integrated meta-controller and shown in the original switching result figure. **a.** Binary stimulus-response class reversals, where the left class becomes the right class, and vice-versa. **b.** Rotations of the binary stimulus-response reward boundaries. **c.** A "squeezing" of the binary stimulus-response reward boundaries, where no reward is given on the new task on the bottom half of the screen, regardless of class shown.

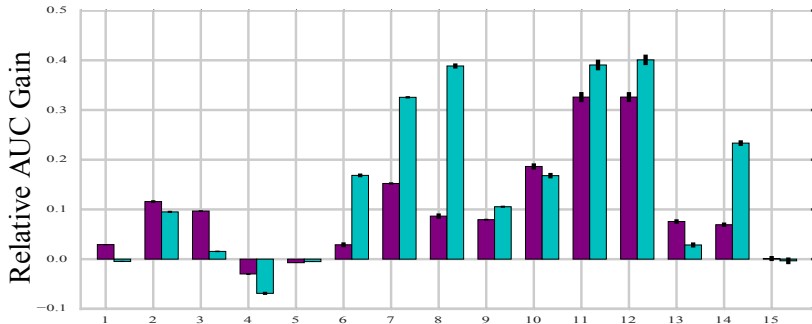

Figure S6: **Targeted controller transform ablation.** Relative AUC Gain for the EMS module over the same switching snearios in the paper, but with the targeted transformations ablated.

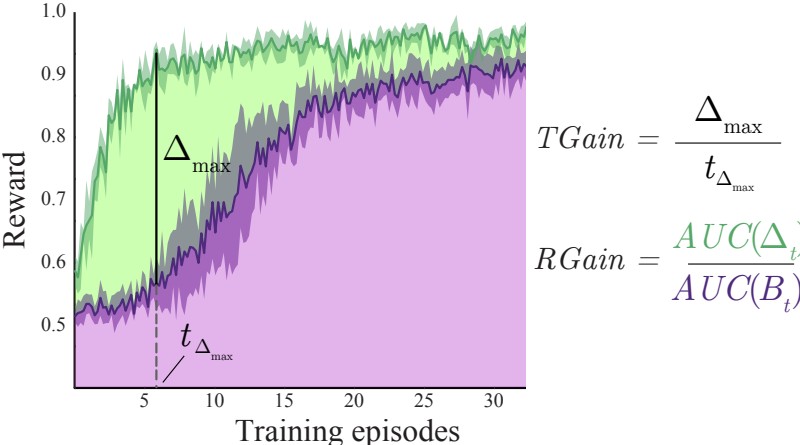

Figure S7: **Illustration of switching performance metrics.** We quantify the switching performance of the dynamic neural controller and task-modules by two metrics: "relative gain in AUC" (ratio of green to purple shaded regions), and "transfer" gain (difference of reward at $T_{\Delta \max}$). Relative AUC measures the overall gain relative to scratch, and the transfer gain measures the speed of transfer. Curve shown is the EMS module with Single-Unit voting method evaluated on a switch from a 4-way MTS task with two randomly moving class templates to a 4-way MTS task with four randomly moving templates.

