# OpenReview forum: "Modular Continual Learning in a Unified Visual Environment"
_ICLR.cc/2018/Conference — Accept (Poster)_

### Official Review · AnonReviewer3 · 2017-11-27
**A interesting scenario with several implications**

**Rating:** 8
**Confidence:** 3

**Review:**

The paper comprises several ideas to study the continual learning problem. First, they show an ad-hoc designed environment, namely the Touchstream environment, in which both inputs and actions are represented in a huge space: as it happens with humans – for example when they are using a touch screen – the resolution of the input space, i.e. the images, is at least big as the resolution of the action space, i.e. where you click on the screen. This environment introduces the interesting problem of a direct mapping between input and actions. Second, they introduce an algorithm to solve this mapping problem in the Touchstream space. Specifically, the ReMaP algorithm learns to solve typical neuroscience tasks, by optimizing a computational module that facilitates the mapping in this space. The Early Bottleneck Multiplicative Symmetric (EMS) module extends the types of computation you might need to solve the tasks in the Touchstream space. Third, the authors introduce another module to learn how to switch from task to task in a dynamical way.
The main concern with this paper is about its length. While the conference does not provide any limits in terms of number of pages, the 13 pages for the main text plus other 8 for the supplementary material is probably too much. I am wondering if the paper just contains too much information for a single conference publication.
As a consequence, either the learning of the single tasks and the task switching experiments could have been addressed with further details. In case of single task learning, the tasks are relatively simple where k_beta = 1 (memory) and k_f = 2 (prediction) are sufficient to solve the tasks. It would have been interesting to evaluate the algorithm on more complex tasks, and how these parameters affect the learning. In case of the task switching a more interesting question would be how the network learns a sequence of tasks (more than one switch).
Overall, the work is interesting, well described and the results are consistent.
Some questions:
- the paper starts with clear inspiration from Neuroscience, but nothing has been said on the biological plausibility of the ReMap, EMS and the recurrent neural voting;
- animals usually learn in continuous-time, thus such tasks usually involve a time component, while this work is designed in a time step framework; could the author argument on that? (starting from Doya 2000)
- specifically, the MTS and the Localization tasks involve working memory, thus a comparison with other working memory with reinforcement learning would make more sense that different degree of ablated modules. (for example Bakker 2002)

Minor:
- Fig 6 does not have letters
- TouchStream is sometimes written as Touchstream

Ref.
Doya, K. (2000). Reinforcement learning in continuous time and space. Neural Computation, 12(1), 219–245.
Bakker, B. (2002). Reinforcement Learning with Long Short-Term Memory. In T. G. Dietterich, S. Becker, & Z. Ghahramani (Eds.), (pp. 1475–1482). Presented at the Advances in Neural Information Processing Systems 14, MIT Press.

---

> ### Author Response · Authors · 2017-12-12
> **Response to Reviewer 3**
>
> > The main concern with this paper is about its length. While the conference does not provide any limits in terms of number of pages, the 13 pages for the main text plus other 8 for the supplementary material is probably too much. I am wondering if the paper just contains too much information for a single conference publication.
>
> Yes, we were originally thinking about breaking the paper into two -- modules and switching. However, the results and conclusions drawn from the EMS/module experiments are useful for understanding the switching section and its associated results. Additionally, due to the novelty of the setup (TouchStream and ReMaP), there would have been substantial overlap between submissions if we were to break it up due to needing to explain each of these components to understand the system as a whole. Overall, these two points convinced us that it was important to create one combined paper with all components elaborated upon, albeit at the expense of having a longer submission.
>
>
> > As a consequence, either the learning of the single tasks and the task switching experiments could have been addressed with further details. In case of single task learning, the tasks are relatively simple where k_beta = 1 (memory) and k_f = 2 (prediction) are sufficient to solve the tasks. It would have been interesting to evaluate the algorithm on more complex tasks, and how these parameters affect the learning. In case of the task switching a more interesting question would be how the network learns a sequence of tasks (more than one switch).
>
> All of these are very good points -- as it is shown in the paper, it is challenging to solve even these tasks and task switching scenarios. Going beyond the shown $k_b$/$k_f$ conditions to more complex temporally-extended tasks and single-task switching scenarios are certainly two of our next steps. We have tried to make it clear in the conclusion that we view these concerns as not completely solved in the present study.
>
> > the paper starts with clear inspiration from Neuroscience, but nothing has been said on the biological plausibility of the ReMap, EMS and the recurrent neural voting;
>
> We ourselves are not yet sure as to whether any of these are indeed biologically plausible mechanisms and thus we refrained from making any claims. It was our primary concern to first design an agent that responds in a behaviorally realistic way to the presented tasks, at least to some extent. Our agent is able to efficiently solve various challenging visual tasks using a touchscreen, and is able to quickly transfer knowledge between what could be two qualitatively distinct tasks. That being said, we are working with experimental collaborators in the neuroscience community to see whether we can indeed find evidence (or lack thereof) for any of these mechanisms or their neural correlates.
>
> > animals usually learn in continuous-time, thus such tasks usually involve a time component, while this work is designed in a time step framework; could the author argument on that? (starting from Doya 2000)
>
> We do not feel particularly strongly about the discrete time framework. Our timesteps were meant to reflect the trial-by-trial structure of typical animals experiments at the moment of decision making. As can be seen in the results, even formulating this problem inside this domain happens to present significant modeling challenges. However, creating a model that functions in continuous time is a good future direction, as there are things such as attention and feedback that are not naturally phrased in the discrete setting.
>
> > specifically, the MTS and the Localization tasks involve working memory, thus a comparison with other working memory with reinforcement learning would make more sense that different degree of ablated modules. (for example Bakker 2002)
>
> The reviewer is correct in implying that the EMS module is not in itself a perfect solution to the problem of working memory (mainly due to fixed $k_b$). Indeed, one of our current focuses is to extend the EMS module motif with more sophisticated memory components and recurrent structures.

---

### Official Review · AnonReviewer2 · 2017-11-30
**An interesting, albeit very dense, paper describing continuous learning over a set of different tasks with a flexible modular architecture and a large state and action space.**

**Rating:** 8
**Confidence:** 2

**Review:**

Reading this paper feels like reading at least two closely-related papers compressed into one, with overflow into the appendix (e.g. one about the EMS module, one about the the recurrent voting, etc).

There were so many aspects/components, that I am not entirely confident I fully understood how they all work together, and in fact I am pretty confident there was at least some part of this that I definitely did not understand. Reading it 5-20 more times would most likely help.

For example, consider the opening example of Section 3. In principle, this kind of example is great, and more of these would be very useful in this paper. This particular one raises a few questions:
-Eq 5 makes it so that $(W \Psi)$ and $(a_x)$ need to be positive or negative together.  Why use ReLu's here at all? Why not just $sign( (W \Psi) a_x) $? Multiplying them will do the same thing, and is much simpler. I am probably missing something here, would like to know what it is... (Or, if the point of the artificial complexity is to give an example of the 3 basic principles, then perhaps point this out, or point out why the simpler version I just suggested would not scale up, etc)
-what exactly, in this example, does $\Psi$ correspond to? In prev discussion, $\Psi$ is always written with subscripts to denote state history (I believe), so this is an opportunity to explain what is different here.
-Nitpick: why is a vector written as $W$? (or rather, what is the point of bold vs non-bold here?)
-a non-bold version of $Psi$, a few lines below, seems to correspond to the 4096 features of VGG's FC6, so I am still not sure what the bold version represents

-The defs/eqns at the beginning of section 3.1 (Sc, CReLu, etc) were slightly hard to follow and I wonder whether there were any typos, e.g. was CReS meant to refer directly to Sc, but used the notation ${ReLu}^2$ instead?

Each of these on its own would be easier to overlook, but there is a compounding effect here for me, as a reader, such that by further on in the paper, I am rather confused.

I also wonder whether any of the elements described, have more "standard" interpretations/notations. For example, my slight confusion propagated further: after above point, I then did not have a clear intuition about $l_i$ in the EMS module. I get that symmetry has been built in, e.g. by the definitions of CReS and CReLu, etc, but I still don't see how it all works together, e.g. are late bottleneck architectures *exactly* the same as MLPs, but where inputs have simply been symmetrized, squared, etc? Nor do I have intuition about multiplicative symmetric interactions between visual features and actions, although I do get the sense that if I were to spend several hours implementing/writing out toy examples, it would clarify it significantly (in fact, I wouldn't be too surprised if it turns out to be fairly straightforward, as in my above comment indicating a seeming equivalence to simply multiplying two terms and taking the resulting sign). If the paper didn't need to be quite as dense, then I would suggest providing more elucidation for the reader, either with intuitions or examples or clearer relationships to more familiar formulations.

Later, I did find that some of the info I *needed* in order to understand the results (e.g. exactly what is meant by a "symmetry ablation", how was that implemented?) was in fact in the appendices (of which there are over 8 pages).

I do wonder how sensitive the performance of the overall system is to some of the details, like, e.g. the low-temp Boltzmann sampling rather than identity function, as described at the end of S2.

My confidence in this review is somewhere between 2 and 3.

The problem is an interesting one, the overall approach makes sense, it is clear the authors have done a very substantial  amount of work, and very diligently so (well-done!), some of the ideas are interesting and seem creative, but I am not sure I understand the glue of the details, and that might be very important here in order to assess it effectively.

---

> ### Author Response · Authors · 2017-12-12
> **Response to Reviewer 2, part 1**
>
> > There were so many aspects/components, that I am not entirely confident I fully understood how they all work together, and in fact I am pretty confident there was at least some part of this that I definitely did not understand. Reading it 5-20 more times would most likely help.
>
> We greatly appreciate the effort that the reviewer took in reading the paper and providing feedback, though we recognize that it is solely our responsibility to present the information as clearly and intuitively as possible. In our revision, we have tried to increase readability and clarity such that the paper can be better understood from a single reading alone.
>
> > For example, consider the opening example of Section 3. In principle, this kind of example is great, and more of these would be very useful in this paper. This particular one raises a few questions:
> - Eq 5 makes it so that $(W \Psi)$ and $(a_x)$ need to be positive or negative together.  Why use ReLu's here at all? Why not just $sign((W \Psi)a_x)$? Multiplying them will do the same thing, and is much simpler. I am probably missing something here, would like to know what it is... (Or, if the point of the artificial complexity is to give an example of the 3 basic principles, then perhaps point this out, or point out why the simpler version I just suggested would not scale up, etc)
>
> The reviewer is correct in the assumption that the additional complexity is to show that starting from a rather specific example, there exist several general concepts which might prove useful in the TouchStream environment for many tasks. As the reviewer implies, it may be the case that there are many equivalent functional forms that can solve the binary SR task. However, by phrasing the solution in terms of standard neural network functions such as an affine transformation followed by a ReLu operation, we found a solution that suggested that a learnable and generic neural network architecture exists as well. This was an important step, because we sought a single module architecture that generalized across all tasks in the TouchStream -- including those without analytical solutions such as MS-COCO. This particular solution happened to display three properties which we then generalized through the CReS nonlinearity and an early bottleneck in the EMS module.
>
> In addition, we have replaced the sign function with the Heaviside function in eq. 5. This does not actually change the value of the equation at all, because the domain is already restricted to nonnegative values, but it makes the mathematical intent clearer.
>
> > what exactly, in this example, does $\Psi$ correspond to? In prev discussion, $\Psi$ is always written with subscripts to denote state history (I believe), so this is an opportunity to explain what is different here.
>
> In this example, it was supposed to refer to the visual encoding at the current timestep. We apologize for any confusion, and have added a time subscript to denote this (see eq. 5).
>
> > Nitpick: why is a vector written as $W$? (or rather, what is the point of bold vs non-bold here?)
>
> Good question. W is intended to represent a generic weight matrix. However, for the purpose of this example, we wanted the visual bottleneck ($W$$\Psi$) to result in a single value indicating which of the two classes is being observed. Technically this means that W is not precisely a vector but a 1 X |$\Psi$| matrix. This has been corrected below eq. 5.
>
> > a non-bold version of $Psi$, a few lines below, seems to correspond to the 4096 features of VGG's FC6, so I am still not sure what the bold version represents
>
> This indeed corresponded to the VGG FC6 feature vector at the current timestep. We have revised for consistency where it appeared in the bulleted "3-general principles" of section 3 and EMS definition of 3.1.

---

> ### Author Response · Authors · 2017-12-12
> **Response to Reviewer 2, part 2**
>
> > The defs/eqns at the beginning of section 3.1 (Sc, CReLu, etc) were slightly hard to follow and I wonder whether there were any typos, e.g. was CReS meant to refer directly to Sc, but used the notation ${ReLu}^2$ instead?
>
> We assume the reviewer was referring to the squaring nonlinearity "Sq" in their comment when they said "Sc" (if not, please correct us). We do not believe that there are any typos in the nonlinearity definitions (although we realized that CReS was defined on te right side of the equation which is inconsistent with Sq & CReLu definitions). CReS as stated is the composition of Sq and CReLu, such that CReS(x) = Sq(CReLu(x)) which concatenates the square of both CReLu components to the CReLu function itself.  In the revised manuscript, we have made the CReS defition consistent with sq/CreLu such that the operator appears on the left side of its definition.
>
> > Each of these on its own would be easier to overlook, but there is a compounding effect here for me, as a reader, such that by further on in the paper, I am rather confused.
>
> We sincerely apologize for any lack of clarity and confusion incurred by the reader. We have put substantial effort into the new revision to correct this. And please feel free to tell us about anything else that you find unclear so that we can improve it.
>
> > I also wonder whether any of the elements described, have more "standard" interpretations/notations. For example, my slight confusion propagated further: after above point, I then did not have a clear intuition about $l_i$ in the EMS module. I get that symmetry has been built in, e.g. by the definitions of CReS and CReLu, etc, but I still don't see how it all works together, e.g. are late bottleneck architectures *exactly* the same as MLPs, but where inputs have simply been symmetrized, squared, etc? Nor do I have intuition about multiplicative symmetric interactions between visual features and actions, although I do get the sense that if I were to spend several hours implementing/writing out toy examples, it would clarify it significantly (in fact, I wouldn't be too surprised if it turns out to be fairly straightforward, as in my above comment indicating a seeming equivalence to simply multiplying two terms and taking the resulting sign). If the paper didn't need to be quite as dense, then I would suggest providing more elucidation for the reader, either with intuitions or examples or clearer relationships to more familiar formulations.
>
> As the reviewer stated, a point that we took to heart in the revision was making sure that even though the material needed to be presented concisely, that it did not sacrifice clarity. This involved restating certain components to be more intuitive in e.g. how the intuition-building example relates to the EMS module (see end of section 3). To answer this question specifically: Yes, the Late-bottleneck architecture is the same as a standard MLP, where actions and visual features are directly concatenated as inputs to the first layer of the module (such that the inputs are $\Psi \oplus a$) without any additional preprocessing (see the last paragraph of section 3.1). Any multiplications and symmetries exist only as a result of the various hidden-layer nonlinearities used in the study. In the main text we show only a "fully-ablated" late bottleneck which uses only ReLu nonlinearities but in the Supplement we also show the case for a late bottleneck using CReLu and hence preserves the symmetry.
>
>
> Although some of the tasks we present do indeed have analytical solutions of the form of eq. 5, the main point we wanted to make was that the three concepts that arise from this example can be generalized across tasks.
>
>
> > Later, I did find that some of the info I *needed* in order to understand the results (e.g. exactly what is meant by a "symmetry ablation", how was that implemented?) was in fact in the appendices (of which there are over 8 pages).
>
> Yes, in the revision process we did catch the fact that it wasn't entirely clear what we meant by the symmetry ablation (replacing CReLu with ReLu or CReS with just Sq(ReLu) for the main paper, or other standard nonlinearities in the Supplement). This has been clarified inside paragraph two of the "Efficiency of the EMS Module" subsection of 3.2.
>
> > I do wonder how sensitive the performance of the overall system is to some of the details, like, e.g. the low-temp Boltzmann sampling rather than identity function, as described at the end of S2.
>
> We have run experiments that indicate that the final performance of any one module on any one task are only somewhat sensitive to changes in ReMaP sampling policy. However, the rank order of results between modules does not change for any task investigated.

---

### Official Review · AnonReviewer1 · 2017-11-30
**Complex network using heuristic structure, state representations, and action selection for solving tasks inspired by psychology**

**Rating:** 6
**Confidence:** 2

**Review:**

The authors propose a kind of framework for learning to solve elemental tasks and then learning task switching in a multitask scenario. The individual tasks are inspired by a number of psychological tasks. Specifically, the authors use a pretrained convnet as raw statespace encoding together with previous actions and learn through stochastic optimization to predict future rewards for different actions. These constitute encapsulated modules for individual tasks. The authors test a number of different ways to construct the state representations as inputs to these module and report results from extensive simulations evaluating them. The policy is obtained through a heuristic, selecting actions with highest reward prediction variance across multiple steps of lookahead. Finally, two variants of networks are presented and evaluated, which have the purpose of selecting the appropriate module when a signal is provided to the system that a new task is starting.

I find it particularly difficult to evaluate this manuscript. The presented simulation results are based on the described system, which is very complex and contains several, non-standard components and heuristic algorithms.

It would be good to motivate the action selection a bit further. E.g., the authors state that actions are sampled proportionally to the reward predictions and assure properties that are not necessarily intuitive, e.g. that a few reward in the future can should be equated to action values. It is also not clear under which conditions the proposed sampling of actions and the voting results in reward maximization. No statements are made on this other that empirically this worked best.

Is the integral in eq. 1 appropriate or should it be a finite sum?

It seems, that the narrowing of the spatial distribution relative to an \epsilon -greedy policy would highly depend on the actual reward landscape, no? Is the maximum variance as well suited for exploration as for exploitation and reward maximization?

What I find a bit worrisome is the ease with which the manuscript switches between  “inspiration” from psychology and neuroscience to plausibility of proposing algorithms to reinterpreting aimpoints as “salience” and feature extraction as “physical structure”. This necessarily introduces a number of

Overall, I am not sure what I have learned with this paper. Is this about learning psychological tasks? New exploration policies? Arbitration in mixtures of experts? Or is the goal to engineer a network that can solve tasks that cannot be solved otherwise? I am a bit lost.


Minor points:
“can from”

---

> ### Author Response · Authors · 2017-12-12
> **Response to Reviewer 1**
>
> > It would be good to motivate the action selection a bit further. E.g., the authors state that actions are sampled proportionally to the reward predictions and assure properties that are not necessarily intuitive, e.g. that a few reward in the future can should be equated to action values. It is also not clear under which conditions the proposed sampling of actions and the voting results in reward maximization. No statements are made on this other that empirically this worked best.
>
> The motivation behind the action selection strategy was one of the points we wished to further clarify in the revision. We agree that we do not offer any theoretical guarantees on the conditions that lead to optimal solutions. Such a theory might be possible, and we believe that in principle it is highly correlated to the geometry of the reward maps. Moving forward, it is certainly on our to-do list to formally prove such a theory. As it stands, the motivation is mostly a heuristic based both on intuition and what works well in practice. However, in our revised submission, we have attempted to provide a clearer intuition as to why it seems to work well in the presented study (see paragraph in section 2.1 beginning "The sampling procedure...").
>
> > It seems, that the narrowing of the spatial distribution relative to an \epsilon -greedy policy would highly depend on the actual reward landscape, no? Is the maximum variance as well suited for exploration as for exploitation and reward maximization?
>
> This is an interesting question, and is related to the previous comment. Indeed, the reviewer is probably correct in the assumption that the utility of any action selection strategy is highly dependent upon the true reward function.
> Our claim is that realistic problems in the physical world usually have similar spatial reward structure to those presented here in that they are typically locally homogeneous. In such cases, an epsilon-greedy policy might focus on only a single point-estimate rather than discovering the entire local physical structure. The revision attempts to clarify how the Maximum-Variance policy works under such conditions, and why it is beneficial for balancing between exploration (choosing something with high uncertainty/variance to learn more about it) and exploitation (choosing something that has a large localized peak in an otherwise uniformly low reward map).
>
> > Is the integral in eq. 1 appropriate or should it be a finite sum?
>
> We formalized the ReMaP section to reflect the general case of a continuous action and reward space, but the reviewer is correct that any particular implementation numerical integration must indeed be approximated by a finite sum.
>
> > What I find a bit worrisome is the ease with which the manuscript switches between  “inspiration” from psychology and neuroscience to plausibility of proposing algorithms to reinterpreting aimpoints as “salience” and feature extraction as “physical structure”. This necessarily introduces a number of
>
> It looks like there is a missing sentence or two for this question. If the reviewer would like to clarify, we're happy to answer afterwards.
>
> > Overall, I am not sure what I have learned with this paper. Is this about learning psychological tasks? New exploration policies? Arbitration in mixtures of experts? Or is the goal to engineer a network that can solve tasks that cannot be solved otherwise? I am a bit lost.
>
> It is in fact about all of these things: our goal was to build and describe new exploration policies and architectural structures (modules and controllers) so as to engineer an agent that can solve and switch between behavorially interesting tasks that otherwise would be difficult to solve efficiently or indeed at all. This naturally involved several things which the reviewer mentioned in their synopsis of the work. First we needed to develop a framework (The TouchStream) to allow for a common phrasing of many interesting and behavorially relevant visual tasks. Next we needed an algorithm (ReMaP) that could explore and solve a large action space such as the TouchStream. Using ReMaP, we still needed to find a neural readout motif (The EMS module) that could learn tasks as efficiently as possible in the TouchStream while remaining lightweight. Finally, we needed a method of reusing our knowledge (Voting Controller) of old tasks for solving new tasks which might be quite qualitatively distinct. The end result of this is what we believe to be a coherent set of results, but we agree that are lot for one paper.

---

### Author Response · Authors · 2017-12-12
**General response to reviewers**

First and foremost, we wish to thank all of the reviewers for taking the time to read and comment on the paper. Across all three reviewers, it appears that the main concern for the submission is its clarity of explanation. Indeed, following submission we realized that there were several places in the paper that could benefit from rewording to improve readability and reader comprehension. As some of the reviewers noted, since the system contains many complex and interacting components, we needed to strike a balance between brevity and clarity. We have revised the manuscript in an effort to strike this balance better than the original submission.

Most of the reviewer comments focused on sections 2 and 3, and in these sections we've made a series of relatively minor changes to improve clarity at the sentence and wording level. We have attempted to go through the individual comments of the reviewers and use them to systematically improve the paper.

We also made more substantial changes in the exposition of the Neural Voting controller in section 4.1. While none of the reviewers commented on this section directly, we felt in retrospect that it was much less clear than we would have liked. We have improved the description of the mathematical formalism to use standard notation whenever possible. In addition, we added a diagram (Fig. 6) to illustrate how the voting controller works.

From a substantive point of view, we made one small addition to the paper. Specifically, we added one control experiment for the switching section (Figure 7: Dynamic Neural Voting quickly corrects for “no-switch” switches). We added this control to ensure that the controller rejects a new module if it is unnecessary (i.e. the case where a 'new' task and previously learned task are identical).

If there are any remaining comments, questions, or confusions that we haven't addressed either in our revised mauscript or the responses below, we would be happy to respond to those throughout the course of the rebuttal period.

---

### Decision · Program_Chairs · 2018-01-29
**ICLR 2018 Conference Acceptance Decision**

**Decision:**

Accept (Poster)

**Comment:**

Important problem (modular continual RL) and novel contributions. The initial submission was judged to be a little dense and hard to read, but the authors have been responsive in responding and updating the paper. I support accepting this paper.